# CANMI: Causal Discovery under Nonstationary Missingness Mechanisms

## Abstract

Causal discovery from time series data is a typical and fundamental problem across various domains. In real-world scenarios, these data often have missing values occurring under different mechanisms, which limits the applicability of most existing approaches, especially when the missing values do not occur randomly due to the influence of other variables. This challenge is further exacerbated when missingness mechanisms also depend on nonstationarity in time series data. In this paper, we propose CANMI, a novel constraint-based approach designed for **CA**usal discovery under **N**onstationary **MI**ssingness mechanisms. Our proposed method can recover the causal structure using only observed data with different missingness mechanisms, including missing not at random (MNAR). Furthermore, we prove the identifiability of the direct causes of missingness and reveal a formula for recovering the data distribution from nonstationary data with missing values. Extensive experiments on both synthetic and real-world datasets demonstrated that our proposed model outperforms state-of-the-art approaches for causal discovery across various evaluation metrics even under substantial missingness. Our source codes are available at https://anonymous.4open.science/r/CANMI-0CDD.

## 1 Introduction

At the heart of comprehending complex systems lies the task of causal discovery – the process of inferring causal relationships from observational data Pearl (2009), and it is utilized across various types of domains, including climate science Kotz et al. (2024), health care Snyder-Mackler et al. (2020); Shen et al. (2020), and neuroscience Lee et al. (2010); Tu et al. (2019), where mechanistic understanding guides prediction and decision-making Aglietti et al. (2020). Randomized controlled trials (RCTs) Fisher (1935) are generally the gold standard for identification of causal relations, but are largely unavailable due to cost and ethical constraints. Therefore, for decades, many researchers have attempted to discover causal relationships purely from observations Glymour et al. (2019); Vowels et al. (2023); Hiremath et al. (2024).

A further challenge with causal discovery methods is the presence of missing values, which may compromise inference accuracy. With the recent development of sensing technologies and the spread of digital survey infrastructures, large amounts of data can be collected and utilized to enhance performance. Nevertheless, regarding real-world scenarios, it is inevitable that such data will have missing values due to various factors, such as sampling drops and deliberate non-responses, making it increasingly important to develop methods capable of handling observational data with missing values Little & Rubin (1987); Zhou et al. (2019); Gao et al. (2022); Obata et al. (2024). Importantly, we must recognize that missing values are often caused by specific variables, rather than occurring randomly. This dependence structure is referred to as the missingness mechanism. Missingness mechanisms can be categorized into three types, i.e., missing completely at random (MCAR), missing at random (MAR), and missing not at random (MNAR) Rubin (1975). When data are MCAR, it is sufficient to perform a list-wise deletion that simply drops the samples with missing values for accurate causal discovery. In contrast, under MAR or MNAR, ignoring the causes of missing values biases the outcome, leading to the unintentional detection of spurious edges and the omission of correct ones.

Nevertheless, these issues are more severe in time series data, where the missingness mechanisms may depend not only on other variables but also on time. A notable characteristic of time series

Table 1: Capabilities of approaches.

| | Constraint | | | Score | | | FCM | | | |
| --- | --- | --- | --- | --- | --- | --- | --- | --- | --- | --- |
| | PC | CD-NOD | MVPC | NOTEARS | GGES | SpaceTime | LiNGAM | PNL | MissDAG | CANMI |
| Causal discovery | ✓ | ✓ | ✓ | ✓ | ✓ | ✓ | ✓ | ✓ | ✓ | ✓ |
| Nonlinear | ✗ | ✓ | ✓ | ✗ | ✓ | ✓ | ✓ | ✗ | ✓ | ✓ |
| Nonstationary data | ✗ | ✓ | ✗ | ✗ | ✗ | ✗ | ✓ | ✗ | ✗ | ✓ |
| Missing completely at random | ✗ | ✗ | ✓ | ✗ | ✗ | ✓ | ✗ | ✗ | ✗ | ✓ |
| Missing not at random | ✗ | ✗ | ✓ | ✗ | ✗ | ✗ | ✗ | ✗ | ✗ | ✓ |
| Nonstationary missingness mechanisms | ✗ | ✗ | ✗ | ✗ | ✗ | ✗ | ✗ | ✗ | ✗ | ✓ |

data is nonstationarity, meaning that the statistical properties of the observed data change over time. Generally, nonstationarity poses additional challenges in the modeling and processing of time series data, and the same holds true for causal discovery. Specifically, this nonstationarity is not limited to the observed data themselves, but may also extend to the missingness mechanisms. It is crucial to focus on this situation if we are to accurately discover causal relationships in more complex scenarios that reflect real-world conditions. For example, with sensor networks during the cold season, machine malfunctions tend to occur due to seasonal factors such as battery failures and condensation-induced short circuits. Without capturing these temporal changes in missingness mechanisms, extraneous edges may be produced unintentionally. Here, we refer to such missingness mechanisms where missing values are caused by both other variables and nonstationarity as nonstationary missingness mechanisms. So, *how can we discover causal relationships from the partially observed data with nonstationary missingness mechanisms?*

A fundamental difficulty in addressing this problem is that we only have access to partially observed data whose distribution is distorted by both nonstationarity and non-random missingness, relative to the complete data distribution. As a result, standard conditional independence (CI) tests applied directly to the observed data may generate spurious edges. To overcome this challenge, we first establish a set of theoretical results, including a recoverability formula for the complete data distribution under nonstationary missingness (Theorem 1). Building on these theoretical foundations, we propose a novel constraint-based algorithm called CANMI designed for **CA**usal discovery under **N**onstationary **MI**ssingness mechanisms. Our algorithm achieves consistent causal discovery by using the theoretical condition in Proposition 4 to detect CI relations that may be artifacts of nonstationary missingness mechanisms, and then correcting corresponding edges through reconstruction of the complete data distribution followed by importance resampling.

**Contributions.** Our method has the following desired contributions:

- Our proposed method can discover causal relationships purely from partially observed data under nonstationary missingness mechanisms, that is, the missing values do not occur randomly caused by both other variables and nonstationarity.

- We provide theoretical results showing that our proposed algorithm correctly discovers the causal structure in our settings. In particular, we prove that it can identify the missingness mechanisms and recover the data distribution required for unbiased CI tests by using only observed data with missing values.

- Extensive experiments on both synthetic and real-world datasets showed that CANMI accurately discovers causal relationships in terms of various types of evaluation metrics and achieves consistent performance even in the presence of substantial missing values, whereas baseline methods exhibited degraded accuracy as the missingness increased.

In addition, Table 1 summarizes six advantages of CANMI. We provide an extensive discussion of related works in Appendix B.

**Outline.** The remainder of this paper is organized in a conventional format. After the introduction, we present preliminaries in Section 2. Next, we theoretically analyze our proposed method and explain our algorithms to identify a causal structure in Section 3. We then provide our experimental results in Section 4, followed by a conclusion in Section 5.

## 2 PRELIMINARIES

We begin by stating the preliminaries for our work. We mainly follow the longitudinal causal discovery setting found in previous research related to our work. We aim to discover causal relationships from observations, so we need to formulate the true causal structure. We provide the details below.

**Notation.** The main symbols and some basic notation and terminology related to causal modeling and missingness that we use in this paper are given in Appendix A. We use italic uppercase such as $X$ and italic bold uppercase such as $\boldsymbol{X}$ to indicate scalar random variables and multivariate random variables, respectively. In addition, upright bold upper-case letters such as $\mathbf{X}$, bold lower-case letters such as $\mathbf{x}$, and regular lower-case letters such as $x$ denote deterministic matrices, vectors, and scalars, respectively. Let $\boldsymbol{V} = (X_1, \ldots, X_d) \in \mathbb{R}^d$ be a $d$-dimensional vector with a joint distribution $P(\boldsymbol{V})$.

### 2.1 CAUSAL BAYESIAN NETWORK

We consider a class of probabilistic graphical models known as causal Bayesian networks, which represent causal relationships among a set of random variables. A causal Bayesian network is defined by $(\mathcal{G}, P)$, where $\mathcal{G}(\boldsymbol{V}, \boldsymbol{E})$ is a directed acyclic graph (DAG) with a node set $\boldsymbol{V} = (X_1, \ldots, X_d)$ and an edge set $\boldsymbol{E}$, and $P$ is a joint probability distribution over these variables. Specifically, under the causal Markov condition (see Section 2.3 for details), a variable $X_i$ is conditionally independent of its non-descendants given its parents. In other words, according to the graph $\mathcal{G}$, the joint distribution $P(\boldsymbol{V})$ is factorized as follows:

$$P(\boldsymbol{V}) = \prod_i P(X_i \mid \mathrm{Pa}(X_i)), \tag{1}$$

where $\mathrm{Pa}(X_i)$ denotes the parents of $X_i$ and $P(X_i \mid \mathrm{Pa}(X_i))$ encodes how the variable $X_i$ is influenced by its parents and called the *causal mechanism* of $X_i$ Reddy & Balasubramanian (2024); Mameche et al. (2025). As a result, the graph $\mathcal{G}$ determines which conditional independence relations must hold in the joint distribution $P(\boldsymbol{V})$. In a causal interpretation, we say that $X$ is a direct cause of $Y$ within the graph $\mathcal{G}$ if and only if $X \in \mathrm{Pa}(Y)$. In summary, if we assume that observations are generated according to the factorization implied by a causal Bayesian network, the process of identifying an underlying graph structure $\mathcal{G}$ from the observed joint distribution $P$ is referred to as causal discovery.

### 2.2 MISSINGNESS GRAPH

To consider formalizing the observed data with missing values, we utilize the notation of the missingness graph (m-graph, for short) Mohan et al. (2013). Let $\mathcal{G}(\mathbb{V}, \boldsymbol{E})$ be an m-graph, which is also a directed acyclic graph, where $\mathbb{V}$ is composed of $\mathbb{V} = \boldsymbol{V} \cup \boldsymbol{V}^* \cup \boldsymbol{R}$. $\boldsymbol{V}$ is the set of observable nodes and is partitioned into a set of fully observed variables $\boldsymbol{V}^o$ and a set of partially observed variables $\boldsymbol{V}^m$. Regarding the partially observed variable $X_i \in \boldsymbol{V}^m$, let $X_i^* \in \boldsymbol{V}^*$ be a proxy variable that corresponds to the actual observed value if available, and takes a missing-entry value (similar to `null`) otherwise. In addition, since we need to represent the status of missingness of the value of the proxy variable $X_i^*$, $R_i \in \boldsymbol{R}$ is introduced and referred to as the *missingness indicator*. Specifically, $R_i = 1$ denotes that the corresponding entry is missing, while $R_i = 0$ indicates that the corresponding entry is observed and the proxy variable $X_i^*$ takes the value of $X_i$.

### 2.3 ASSUMPTIONS

Here, we provide the assumptions used throughout the following sections.

**Assumption 1** (Pseudo causal sufficiency)**.** *We assume that all the possible confounders can be written as smooth functions of the time index. It follows that at each time instance, the values of these confounders are fixed.*

**Assumption 2** (Causal Markov condition). *Each variable $X$ is independent of all its non-descendants, given its parents $\mathrm{Pa}(X)$ in the graph $\mathcal{G}$, i.e., $X \perp\!\!\!\perp \boldsymbol{V} \setminus \mathrm{Pa}(X) \cup \mathrm{De}(X) \cup \{X\} \mid \mathrm{Pa}(X)$.*

**Assumption 3** (Faithfulness). *Let $P$ be a probability distribution generated by the graph $\mathcal{G}$. $P$ is faithful to $\mathcal{G}$ if each conditional independence relationship in $P$ implies d-separation in $\mathcal{G}$.*

**Assumption 4** (No child node for missingness indicators). *No missingness indicator can be the cause of any variable, i.e., for any $R$, we have $\mathrm{Ch}(R) = \emptyset$.*

**Assumption 5** (Faithful observability). *Any conditional independence relation in the observed data also holds in the unobserved data, i.e., for any $X, Y \in \boldsymbol{V}$, it holds that $X \perp\!\!\!\perp Y \mid \{\mathbf{S}, \boldsymbol{R} = \mathbf{0}\} \iff X \perp\!\!\!\perp Y \mid \{\mathbf{S}, \boldsymbol{R} = \mathbf{1}\}$, where $\boldsymbol{R}$ is the missingness indicator set $\{R_X, R_Y, \boldsymbol{R_S}\}$. Specifically, $\boldsymbol{R} = \mathbf{0}$ indicates that every $R \in \boldsymbol{R}$ takes the value zero, while $\boldsymbol{R} = \mathbf{1}$ indicates that there exists at least $R \in \boldsymbol{R}$ taking the value one.*

**Assumption 6** (No causal interactions between missingness indicators). *No missingness indicator can be a deterministic function of any other missingness indicator.*

**Assumption 7** (No self-masking missingness). *Self-masking missingness shows that there is a missingness indicator that is caused by the corresponding variable. For example, for a variable $X$, this is described by $X \to R_X$. This assumption indicates that there is no such edge in the graph $\mathcal{G}$.*

Note that Assumptions 1-3 are conventional and commonly utilized in most existing research on causal discovery. In contrast, Assumptions 4-7 are required in order to handle missing values with different missingness mechanisms. We provide detailed intuitive descriptions that these assumptions are required and minimal in Appendix C.

## 3 Proposed Method: CANMI

In this section, we present CANMI, a novel approach for causal discovery under nonstationary missingness mechanisms. Our proposed method can handle changes in the underlying data generating process over time and the occurrence of missing values under different mechanisms.

### 3.1 Problem Definition

We aim to identify causal relationships from partially observed data whose distribution changes over time. Let $\mathcal{G}(\boldsymbol{V}, \boldsymbol{E})$ be the underlying contemporaneous causal graph over the complete variables $\boldsymbol{V} = (X_1, \ldots, X_d) \in \mathbb{R}^d$, and let $\boldsymbol{R} \in \{0, 1\}^d$ be the missingness indicators. Note that our approach can be naturally generalized to incorporate time-lagged dependencies, analogous to how a constraint-based approach was adapted to handle time series data Chu & Glymour (2008). We consider a sequence of samples indexed by time $t \in \{1, \ldots, N\}$. At time point $t$, the complete data vector $\boldsymbol{V}_t$ and the missingness indicators $\boldsymbol{R}_t$ are generated from a distribution $P_t(\boldsymbol{V}, \boldsymbol{R})$. Here, we focus on nonstationary settings; that is, there exist $t_1 \neq t_2$ such that $P_{t_1}(\boldsymbol{V}, \boldsymbol{R}) \neq P_{t_2}(\boldsymbol{V}, \boldsymbol{R})$. This distributional change is driven by latent factors $\boldsymbol{U}$ that represent slowly changing environments (e.g., seasonal effects and user behavior trends). Both the variables in $\boldsymbol{V}$ and the missingness indicators $\boldsymbol{R}$ depend on $\boldsymbol{U}$. This formulation assumes that the causal graph $\mathcal{G}$ is time-invariant. Also, we cannot observe the complete variables $\boldsymbol{V}$ and the latent factors $\boldsymbol{U}$ in our settings. Instead, we can use only the proxy variables $\boldsymbol{V}^*$ and the corresponding missingness indicators $\boldsymbol{R}$. Consequently, we need to solve the following discrepancy:

**Proposition 1.** *Suppose that for $X, Y \in \boldsymbol{V}$, $\mathbf{S} \subseteq \boldsymbol{V}$, and latent factors $\boldsymbol{U}$ that drive the nonstationary changes, even if it holds that $X \perp\!\!\!\perp Y \mid \mathbf{S} \cup \boldsymbol{U}$, it does not necessarily hold that $X \perp\!\!\!\perp Y \mid \mathbf{S} \cup \{R_X = 0, R_Y = 0, \boldsymbol{R_S} = \mathbf{0}\}$.*

Intuitively, the reason for this discrepancy is that we can use only partially observed samples, and latent nonstationary factors $\boldsymbol{U}$ influence not only the variables $\boldsymbol{V}$ but also the missingness indicators $\boldsymbol{R}$. Therefore, conditioning on missingness indicators acts like conditioning on a collider influenced by these time-varying latent factors, which can introduce spurious dependencies between $X$ and $Y$ that were absent under conditioning on $\mathbf{S} \cup \boldsymbol{U}$. Resolving this discrepancy is non-trivial, as it arises in complex scenarios where the distributional nonstationarity and the missingness mechanisms are mutually dependent through both the latent factors and conditioning on the missingness indicators. In summary, the problem addressed in this paper is as follows:

*Given partially observed data $\mathcal{D} = \{(\boldsymbol{V}_t^*, \boldsymbol{R}_t)\}_{t=1}^N$ generated as above, where the complete data distribution $P_t(\boldsymbol{V}, \boldsymbol{R})$ may vary over time due to latent factors $\boldsymbol{U}$, Identify a causal graph $\mathcal{G}(\boldsymbol{V}, \boldsymbol{E})$ under Assumptions 1-7.*

## 3.2 THEORETICAL ANALYSIS

In this section, we provide theoretical results that validate the proposed algorithm presented in Section 3.3. Before turning to the main results, we introduce the time index variable $T$ as a surrogate variable (i.e., it is not part of a causal system) that is instrumental in our analysis. Specifically, we assume that when the causal mechanism governing the variable $X$ changes over time, such a change can be attributed to an unobserved confounder denoted as $u(T)$. Furthermore, all of the proofs for our theoretical results in this section are provided in Appendix D. First, we discuss the identifiability of the causal relationships involving the missing indicators.

**Proposition 2.** *Under Assumptions 1-7, the direct causes of missingness indicators are identifiable.*

As mentioned briefly in the previous section, nonstationarity gives rise to an apparent presence of unobserved confounders, which may generate spurious causal relations. However, this theoretical analysis indicates that the parents responsible for the missingness of a variable, including the unobserved confounders, are identifiable.

**Proposition 3.** *Under Assumptions 1-7, for any $X, Y \in \boldsymbol{V}$ and $\mathbf{S} \subseteq \boldsymbol{V} \cup \{T\} \setminus \{X, Y\}$, if it holds that $X \per\!\!\!\perp Y \mid \mathbf{S} \cup \{R_X = 0, R_Y = 0, \boldsymbol{R_S} = \mathbf{0}\}$, it also holds that $X \per\!\!\!\perp Y \mid \mathbf{S}$, where $T$ is the time index and $\boldsymbol{R_S} = \mathbf{0}$ means that every $R \in \boldsymbol{R_S}$ takes the value zero.*

Proposition 3 implies that if a CI relation is obtained using only the partially observed data and the time index $T$, then the same CI relation also holds in the full data distribution. This analysis justifies the use of partially observed data and the time index $T$ for CI tests before recovering the data distribution in our algorithm (Step 3). Next, we discuss the conditions where extraneous conditional relations occur in the observed data due to missingness.

**Proposition 4.** *Suppose that $X$ and $Y$ are not adjacent in a true m-graph, and that for any $\mathbf{S} \subseteq \boldsymbol{V} \cup \{T\} \setminus \{X, Y\}$ such that $X \per\!\!\!\perp Y \mid \mathbf{S}$, it holds $X \not\per\!\!\!\perp Y \mid \mathbf{S} \cup \{R_X = 0, R_Y = 0, \mathbf{R_S} = \mathbf{0}\}$. Then, under Assumptions 1-7, for at least one variable $Z \in \{X\} \cup \{Y\} \cup \mathbf{S}$, the missingness indicator $R_Z$ is either the direct common effect or a descendant of the direct common effect of $X$ and $Y$.*

To achieve accurate causal discovery, we need to clarify whether potential extraneous CI relations as mentioned in Proposition 4 are truly present in the underlying true m-graph. We tackle this problem by applying density ratio weighting to adjust for the missingness mechanism.

**Theorem 1** (Recoverability of data distribution). *Under Assumptions 1-7, given the parents of each missingness indicator $\mathrm{Pa}(R_i)$, the joint distribution $P(\boldsymbol{V})$ is recoverable, and we then have*

$$P(\boldsymbol{V}) = \frac{P(\boldsymbol{R} = \mathbf{0}, \boldsymbol{V})}{\prod_i P(R_i = 0 \mid \mathrm{Pa}^+(R_i), \boldsymbol{R}_{\mathrm{Pa}(R_i)} = 0)} \tag{2}$$

$$= \frac{1}{Z} P(\boldsymbol{V} \mid \boldsymbol{R} = \mathbf{0}) \prod_i \omega_{\mathrm{Pa}(R_i)} \tag{3}$$

*where*

$$\mathrm{Pa}^+(R_i) = \mathrm{Pa}(R_i) \cup \{T\} \qquad \qquad \textit{(time-augmented parents)},$$

$$Z = \frac{\prod_i P(R_i = 0 \mid \boldsymbol{R}_{\mathrm{Pa}(R_i)} = \mathbf{0})}{P(\boldsymbol{R} = \mathbf{0})} \qquad \qquad \textit{(normalizing constant)},$$

$$\omega_{\mathrm{Pa}(R_i)} = \frac{P(\mathrm{Pa}^+(R_i) \mid \mathbf{R}_{\mathrm{Pa}(R_i)} = \mathbf{0})}{P(\mathrm{Pa}^+(R_i) \mid R_i = 0, \mathbf{R}_{\mathrm{Pa}(R_i)} = \mathbf{0})} \qquad \qquad \textit{(density ratio weights)}.$$

Also, it is known that typical constraint-based methods are order-dependent, in other words, their output depends on the order of CI tests. However, our proposed algorithm is fully order-independent.

**Theorem 2** (Order independence). *Under Assumptions 1-7, the skeleton result from CANMI algorithm is independent of the order of variables $(X_1, \ldots, X_d)$.*

---

**Algorithm 1** CANMI

---

**Input:** Partially observed data $\mathbf{X} = \{\mathbf{x}^{(t)}\}_{t=1}^N$ with length $N$, Significance threshold $\alpha$
**Output:** Causal structure $\mathcal{G}$
  1: Step 1: Detecting direct effects of changing causal mechanisms by employing CI tests between each variable and the time index.
  2: Step 2. Identifying variables that influence missingness indicators by employing CI tests between each variable and the corresponding missingness indicators.
  3: Step 3. Searching for the causal skeleton of $\mathcal{G}$ in the observations $\boldsymbol{X}$, discarding any records with missing values for variables involved in each CI test.
  4: Step 4. Removing potential extraneous edges using recovered joint distribution according to Theorem 1.
  5: Step 5. Determining the orientation of as many edges in $\mathcal{G}$ as possible by applying Meek's orientation rules.

---

### 3.3 ALGORITHM

In this section, we describe in detail the CANMI algorithm for identifying the causal structure under nonstationary missingness mechanisms. The algorithm integrates conditional independence (CI) tests with time-varying proxy structures, while mitigating the spurious dependencies induced by influences of other variables and nonstationarity. Algorithm 1 provides an overview of our proposed algorithm. We provide a step-by-step description below:

Step 1: Detecting changing causal mechanisms

We begin by identifying variables whose causal mechanisms change over time. For each variable $X \in \boldsymbol{V}$, we perform CI tests between $X$ and the time index $T$ conditioned on subsets of $\boldsymbol{V} \setminus \{X\}$. If there is a subset $\mathbf{S} \subseteq \boldsymbol{V} \setminus \{X\}$ such that $X \perp\!\!\!\perp T \mid \mathbf{S}$, then we consider the corresponding causal mechanism of $X$ to be time-invariant; otherwise, we regard one of $X$ as time-variant.

Step 2: Identifying parents of missingness indicators

Next, for each missingness indicator $R_X$, we detect its potential causes by CI tests between $R_X$ and each variable $Y \in \boldsymbol{V} \cup \{T\} \setminus \{X\}$, conditioned on subsets of $\boldsymbol{V} \cup \{T\} \setminus \{X, Y\}$. If it finds a subset $\mathbf{S} \subseteq \boldsymbol{V} \cup \{T\} \setminus \{X, Y\}$ such that $R_X \perp\!\!\!\perp Y \mid \mathbf{S}$, it reveals that $Y$ is not a parent of $R_X$. This step identifies direct causes of missingness indicators, including time-dependent causes, in accordance with Proposition 2. The resulting estimates are then used in Step 4 to reconstruct the data distribution.

Step 3: Skeleton search under missingness

We construct the skeleton of the causal graph, using only the partially observed data associated with each CI test. Specifically, for any $X, Y \in \boldsymbol{V}$, we discard samples in which at least one of $X, Y$, or any variable from the conditioning set $\mathbf{S} \subseteq \boldsymbol{V} \cup \{T\} \setminus \{X, Y\}$ used in the CI test are missing. If the two nodes $X, Y$ are determined to be conditionally independent (i.e., $X \perp\!\!\!\perp Y \mid \mathbf{S}$) at this step, it is also satisfied in the true graph by Proposition 3. Note that Proposition 3 provides a sufficient condition only; the converse does not generally hold. Such an issue is addressed in Step 4 via distributional recoverability and reweighting.

Step 4: Pruning extraneous edges

Some edges in the estimated causal skeleton in Step 3 may be artifacts of missingness and nonstationarity in the observed data. To address the issue, we apply the recoverability formula Equation (2) shown in Theorem 1 to reconstruct the joint distribution $P(\boldsymbol{V})$, and employ it to eliminate potential spurious edges. When we perform a CI test between $X$ and $Y$ given $\mathbf{S}$, we only focus on $\boldsymbol{V}_{\text{CI}} = \{X\} \cup \{Y\} \cup \mathbf{S} \cup \mathbf{Z}$ where $\mathbf{Z} = \{\text{Pa}(R_X), \text{Pa}(R_Y), \text{Pa}(R_{\mathbf{S}}), \text{Pa}(R_{\mathbf{Z}})\}$, which increases the number of samples we can use, leading to a more accurate CI test. Specifically, we use kernel density estimation (KDE) Sheather & Jones (1991) to compute each density ratio $\omega_{\text{Pa}(R_i)}$. With the estimated weights as a basis, we perform importance resampling to generate a modified dataset that more faithfully reflects the underlying true distribution.

Step 5: Edge orientation via Meek's rules

Finally, we apply Meek's orientation rules Meek (1995) to direct as many edges as possible, utilizing known v-structures and acyclicity constraints. This procedure ensures consistency with the established skeleton and causal semantics.

We conclude this section by introducing the soundness of our proposed algorithm CANMI built according to our theoretical results.

**Theorem 3** (Soundness of CANMI). *Under Assumptions 1-7, CANMI returns a causal skeleton graph that is exactly consistent with the true causal skeleton.*

## 4 EXPERIMENTS

We conducted extensive experiments to evaluate the performance of CANMI from various perspectives, using both synthetic and real-world datasets. In the main text, we reported the accuracy and robustness of CANMI and validated on the fMRI datasets to demonstrate the applicability of our method. In particular, since it is believed that one of the basic properties of the neural connections is their time-dependence Havlicek et al. (2011), fMRI data is suitable for the evaluation of CANMI. Additional results, including statistical analysis, sensitivity analysis, and computational efficiency, are provided in Appendix E.

### 4.1 EXPERIMENTAL SETUP

Here, we briefly provide the settings we used for our experiments. The reader can find a more detailed description in Appendix E.1.

**Baselines.** We compared our proposed method with the following eight state-of-the-art approaches for causal discovery, comprising (i) two constraint-based methods (CD-NOD Huang et al. (2020), MVPC Tu et al. (2019)), (ii) four score-based methods (SpaceTime Mameche et al. (2025), NOTEARS-MLP Zheng et al. (2019), NOTEARS Zheng et al. (2018), GGES Huang et al. (2018)), (iii) two FCM-based methods (MissDAG Gao et al. (2022), LiNGAM Shimizu et al. (2006)). To better explore the effectiveness of data distribution recovery, we prepared a limited version, namely CANMI-L, which leaves extraneous edges without Step 4. In the following sections, we abbreviate NOTEARS-MLP as NO-MLP for simplicity.

**Evaluation metrics.** For all our experiments, we reported five common scores with which to evaluate the estimated causal dependencies, namely true positive rate (TPR), false positive rate (FPR), false discovery rate (FDR), F1-score (F1), and structural Hamming distance (SHD).

### 4.2 SYNTHETIC DATASETS

First, we quantitatively compared CANMI with its baselines using synthetic datasets whose ground truth is known in advance.

**Data generating process.** First, we explain our synthetic data generating process. We mainly followed the procedure of Liu & Constantinou (2022) and adapted it to align with our problem setting. Specifically, for each variable $X_i$, we generated time series data $\{x_i^{(t)}\}_{t=1}^{N}$ of length $N$ by

$$x_i^{(t)} = \sum_{j \in \text{pa}(i)} b_{i,j} f_i(x_j^{(t)}) + \varepsilon_i^{(t)}.$$

where, $b_{ij}$ is the strength of the causal dependency between $X_i$ and its parents and $\varepsilon_i^{(t)} \sim \mathcal{N}(0, 1)$ is a mutually independent exogenous variable corresponding to $X_i$. Furthermore, $f_i$ denotes a nonlinear deterministic function describing the causal mechanisms of $X_i$ constructed by MLPs, where we consider two hidden layers and each of which has 100 hidden dimensions. A random graph $\mathcal{G}$ was generated from a well-known random graph model, namely Erdös-Rényi (ER) Erdös & Rényi (1960). Given a graph $\mathcal{G}$, we sampled the strengths of the influence $b_{i,j}$ from $\mathcal{U}([-2.0, -0.5] \cup [0.5, 2.0])$. In addition, we needed to validate the effectiveness of our proposed method against the ubiquitous properties of time series, i.e., nonstationarity. Specifically, we randomly selected $50\%$ of the functions $f_i$ as changing causal mechanisms. We changed these functions by adding a nonstationarity driver $c(t) = \sin(2\pi k_c t/N + \phi)$, where $k_c = 2$ is the number of cycles and $\phi$ is a phase that is randomly chosen from $\mathcal{U}(0, 2\pi)$.

Table 2: Causal discovery results in nonlinear settings. We used dimensional synthetics data of length $N = 3000$ generated based on {ER2, ER4} graphs. The best results are in **bold** and the second best are underlined. CANMI consistently outperforms its baselines across various evaluation metrics.

| Metric | | TPR ($\uparrow$) | FPR ($\downarrow$) | FDR ($\downarrow$) | F1 ($\uparrow$) | SHD ($\downarrow$) |
|---|---|---|---|---|---|---|
| LiNGAM | ER2 | $0.161 \pm 0.040$ | $0.464 \pm 0.126$ | $0.903 \pm 0.025$ | $0.121 \pm 0.030$ | $14.0 \pm 4.062$ |
| | ER4 | $0.151 \pm 0.102$ | $0.493 \pm 0.177$ | $0.834 \pm 0.129$ | $0.157 \pm 0.111$ | $19.2 \pm 2.588$ |
| GGES | ER2 | $0.518 \pm 0.176$ | $0.100 \pm 0.016$ | $0.400 \pm 0.064$ | $0.542 \pm 0.088$ | $7.4 \pm 2.408$ |
| | ER4 | $0.357 \pm 0.060$ | $\mathbf{0.036 \pm 0.051}$ | $0.143 \pm 0.202$ | $0.502 \pm 0.092$ | $11.8 \pm 2.387$ |
| NOTEARS | ER2 | $0.609 \pm 0.158$ | $0.200 \pm 0.103$ | $0.286 \pm 0.165$ | $0.649 \pm 0.144$ | $5.8 \pm 2.775$ |
| | ER4 | $0.264 \pm 0.138$ | $0.043 \pm 0.064$ | $0.173 \pm 0.241$ | $0.381 \pm 0.165$ | $12.6 \pm 2.408$ |
| NO-MLP | ER2 | $0.877 \pm 0.081$ | $0.200 \pm 0.103$ | $0.402 \pm 0.118$ | $0.706 \pm 0.093$ | $6.2 \pm 2.775$ |
| | ER4 | $0.615 \pm 0.170$ | $0.157 \pm 0.117$ | $0.276 \pm 0.091$ | $0.651 \pm 0.103$ | $9.6 \pm 2.510$ |
| MVPC | ER2 | $0.905 \pm 0.096$ | $0.050 \pm 0.032$ | $0.152 \pm 0.103$ | $0.869 \pm 0.046$ | $2.2 \pm 0.837$ |
| | ER4 | $0.655 \pm 0.028$ | $0.086 \pm 0.041$ | $0.180 \pm 0.094$ | $0.725 \pm 0.033$ | $7.6 \pm 0.548$ |
| CD-NOD | ER2 | $0.855 \pm 0.055$ | $0.093 \pm 0.041$ | $0.261 \pm 0.111$ | $0.788 \pm 0.066$ | $3.4 \pm 0.894$ |
| | ER4 | $0.765 \pm 0.195$ | $0.071 \pm 0.051$ | $0.141 \pm 0.083$ | $0.802 \pm 0.141$ | $5.0 \pm 3.536$ |
| MissDAG | ER2 | $0.905 \pm 0.096$ | $0.057 \pm 0.041$ | $0.161 \pm 0.097$ | $0.865 \pm 0.049$ | $2.2 \pm 0.837$ |
| | ER4 | $\mathbf{0.770 \pm 0.239}$ | $0.093 \pm 0.032$ | $0.174 \pm 0.043$ | $0.783 \pm 0.161$ | $6.0 \pm 3.536$ |
| SpaceTime | ER2 | $0.429 \pm 0.052$ | $0.321 \pm 0.062$ | $0.707 \pm 0.077$ | $0.346 \pm 0.070$ | $10.2 \pm 2.280$ |
| | ER4 | $0.501 \pm 0.205$ | $0.279 \pm 0.099$ | $0.491 \pm 0.200$ | $0.504 \pm 0.202$ | $11.6 \pm 4.506$ |
| CANMI-L | ER2 | $0.887 \pm 0.151$ | $0.079 \pm 0.043$ | $0.220 \pm 0.098$ | $0.825 \pm 0.103$ | $3.0 \pm 1.717$ |
| | ER4 | $0.726 \pm 0.127$ | $0.071 \pm 0.033$ | $0.140 \pm 0.062$ | $0.783 \pm 0.096$ | $5.2 \pm 2.546$ |
| CANMI | ER2 | $\mathbf{0.922 \pm 0.130}$ | $\mathbf{0.029 \pm 0.016}$ | $\mathbf{0.093 \pm 0.062}$ | $\mathbf{0.910 \pm 0.083}$ | $\mathbf{1.6 \pm 1.517}$ |
| | ER4 | $0.765 \pm 0.160$ | $0.043 \pm 0.047$ | $\mathbf{0.092 \pm 0.115}$ | $\mathbf{0.822 \pm 0.120}$ | $\mathbf{4.6 \pm 2.608}$ |

Next, we added missing values to complete time series data as explained above. First, we randomly selected $50\%$ of the variables as partially observed variables and chose up to 2 parents of their missingness indicators from both complete and partially observed variables for the MNAR mechanism. In addition, we randomly selected $50\%$ of the partially observed variables as time-varying missingness, and we set one of the parents to the time index $T$. The procedure for removing observations from partially observed variables is as follows. If a missingness indicator had no parents, we removed observations with missing probability $p = 0.3$. Alternatively, if the missingness indicator had parents, we removed them with missing probability $p = p_h$ when the parent fell within its most frequently populated bin under a discretization with five bins per dimension. Otherwise, we removed them with missing probability $p = \min(p_h, 0.1)$. In addition, we focus that missingness mechanisms change over time in our work. Thus, we randomly selected $50\%$ of the missingness indicators and added sine wave functions $c(t)$ to them.

**Effectiveness.** We demonstrated the accuracy with which CANMI can discover causal relationships from partially observed data compared with its baselines. Table 2 shows the overall causal discovery results on 8 dimensional synthetic data of length $N = 3000$ generated based on {ER2, ER4} graphs and missing probability $p_h = 0.6$, where the best and second-best levels of performance are shown in **bold** and underlined, respectively. These results show that CANMI outperforms the state-of-the-art baselines by precisely accounting for complicated missingness mechanisms that depend on both other variables and nonstationarity, which is consistent with our theoretical results provided in Section 3.2. Recalling that nonstationarity and missing values may unintentionally produce extraneous CI relations, the substantially lower FPR and FDR achieved by CANMI indicate that it effectively mitigates such artifacts, thereby leading to a more accurate identification of the underlying causal structure. SpaceTime aims to discover the causal relationships from nonstationary time series data, but it implicitly assumes that nonstationary time series data consist of multiple piecewise-stationary segments, which cannot handle smoothly changing causal mechanisms, resulting in decreased discovery accuracy. While MVPC is capable of recovering the data distribution from observed data with missing values, the temporal dependency inherent in the synthetic data can lead to biased and spurious relations. Since the accurate recovery of the data distribution depends on appropriately identifying the parents of each missingness indicator, MVPC exhibits poorer detection

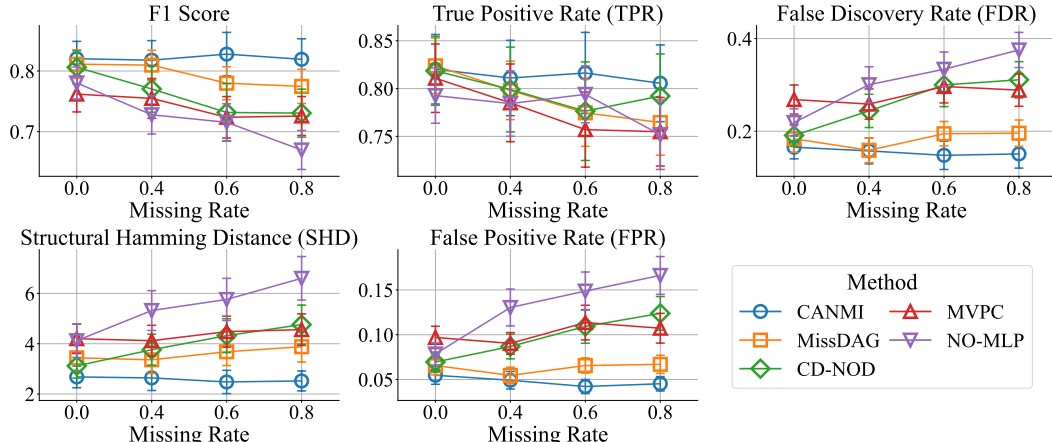

Figure 1: Causal discovery results for synthetic datasets with different missing probabilities $p_h \in \{0.0, 0.4, 0.6, 0.8\}$. CANMI consistently achieved strong discovery performance across any missingness levels while its baselines result in decreased discovery accuracy.

power in such settings, especially when the true graph is relatively dense. Moreover, GGES is comparable to our proposed method CANMI in ER4 from the perspective of the FPR, but CANMI still outperforms it significantly regarding other evaluation metrics. Note that the lowest FPR can be trivially achieved by an empty graph; however, this renders the result meaningless in practice. The approaches that rely solely on the linear causal model (i.e., LiNGAM and NOTEARS) inherently fail to capture nonlinear dependencies. We can see that CANMI-L causes a drop in accuracy. This suggests that the recovery of the data distribution are crucial for reliable causal discovery under nonstationary missingness mechanisms, consistent with the discussion presented in Theorem 2. In particular, FDR and SHD values on ER2 graphs were significantly higher. This is because more extraneous edges were generated due to the sparsity of the true causal graph, and CANMI-L failed to prune them.

**Robustness.** We evaluated the robustness of CANMI compared with competitive baselines (i.e., MissDAG, CO-NOD, MVPC, and NO-MLP) on synthetic datasets spanning a range of missing probabilities and dimensionalities. Figure 1 shows the results obtained with different missing probabilities $p_h \in \{0.0, 0.4, 0.6, 0.8\}$. For brevity, additional results for various dimensionalities are shown in Figure 2 of Appendix E.2. These results indicate that CANMI consistently achieves noteworthy performance in terms of all evaluation metrics, specifically the F1 score and SHD, across various levels of missingness while its baseline methods show substantial degradation as the missing probability $p_h$ increases. This is because none of its baselines have the ability to handle partially observed data, where missing values occur not at random but instead are caused by both other variables and nonstationarity. In particular, we can observe a significant drop in the discovery performance of NO-MLP, which cannot handle both nonstationary data and non-random missingness. Therefore, our proposed method exhibits robustness against high missing probabilities, which indicates that it remains applicable in practice even when missingness is substantial and not at random.

## 4.3 REAL-WORLD DATASETS

We show the applicability of CANMI for real-world fMRI datasets. We utilized the NetSim dataset[1] which describes the connecting dynamics of 15 human brain regions from blood oxygenation level-dependent (BOLD) imaging data Smith et al. (2011). It is commonly used as a benchmark for evaluating temporal causal discovery methods Gong et al. (2023). We introduced missingness by following the same procedure used for synthetic datasets because the NetSim dataset originally has no missing values. Table 3 presents the causal discovery results of CANMI and its competitive baselines on the NetSim dataset with missing values, where the best and second-best levels of performance

---

[1]https://www.fmrib.ox.ac.uk/datasets/netsim/index.html

Table 3: Causal discovery results for the NetSim dataset with missing values across 10 subjects, where the best results are in **bold** and the second best are underlined. CANMI consistently outperforms its baselines in terms of various types of evaluation metrics.

| Metric | TPR ($\uparrow$) | FPR ($\downarrow$) | FDR ($\downarrow$) | F1 ($\uparrow$) | SHD ($\downarrow$) |
|---|---|---|---|---|---|
| NO-MLP | $0.382 \pm 0.049$ | $0.786 \pm 0.053$ | $0.923 \pm 0.010$ | $0.128 \pm 0.017$ | $86.500 \pm 4.913$ |
| MVPC | $0.461 \pm 0.087$ | $\underline{0.030 \pm 0.012}$ | $\underline{0.280 \pm 0.109}$ | $0.561 \pm 0.096$ | $\underline{10.400 \pm 1.823}$ |
| CD-NOD | $\underline{0.509 \pm 0.120}$ | $0.044 \pm 0.022$ | $0.326 \pm 0.128$ | $\underline{0.571 \pm 0.104}$ | $12.500 \pm 2.022$ |
| MissDAG | $0.285 \pm 0.067$ | $0.676 \pm 0.077$ | $0.931 \pm 0.022$ | $0.111 \pm 0.033$ | $77.062 \pm 7.996$ |
| CANMI | $\mathbf{0.520 \pm 0.054}$ | $\mathbf{0.030 \pm 0.008}$ | $\mathbf{0.250 \pm 0.060}$ | $\mathbf{0.613 \pm 0.051}$ | $\mathbf{10.071 \pm 1.346}$ |

are shown in **bold** and underlined, respectively. CANMI consistently outperformed its baselines, meaning that it is practical for real-world scenarios.

## 5 CONCLUSION

Our work investigated the challenges associated with causal discovery under the mechanisms where missing values are caused by both other variables and nonstationarity. To address this difficulty, we presented a novel constraint-based method CANMI, which discovers causal relationships purely from partially observed data with nonstationary missingness mechanisms. Our proposed method achieved our goal by theoretically detecting the direct causes of missingness indicators under nonstationary missingness mechanisms (Proposition 2) and the recovery of the joint distribution required for removing extraneous edges through unbiased CI tests (Theorem 1). Our experimental evaluation on both synthetic and real-world datasets showed that CANMI achieved highly accurate causal discovery compared with multiple state-of-the-art competitors across various types of evaluation metrics.

## ETHICS STATEMENT

In this work, we proposed a novel constraint-based causal discovery method under complicated but ubiquitous missingness mechanisms that depend on both other variables and nonstationarity (i.e., nonstationary missingness mechanisms). Given the scope of our research, we do not anticipate any significant negative societal or ethical consequences arising from our proposed method. Also, we evaluated our method on synthetic and public datasets only; no personally identifiable or sensitive data are used.

## REPRODUCIBILITY STATEMENT

All assumptions used in this paper and proofs of our theoretical results are provided in Section 2.3 and Appendix D, respectively. Our source codes that we used in the experiments are available at https://anonymous.4open.science/r/CANMI-0CDD, which will be made public after the review process. We described the experimental setup including computing infrastructure and implementation details in Section 4.1 and Appendix E.1.

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

APPENDIX

# A  NOTATION AND TERMINOLOGY

## A.1  CAUSAL BAYESIAN NETWORK

We summarize the causal relations between two nodes used in the longitudinal causal discovery framework.

- **Children**: let $\mathrm{Ch}(X)$ be causal children of $X$, i.e., for every $X_j \in \mathrm{Ch}(X_i)$, we write $X_i \to X_j$.
- **Parents**: let $\mathrm{Pa}(X)$ be causal parents of $X$, i.e., for every $X_j \in \mathrm{Pa}(X_i)$, we write $X_j \to X_i$.
- **Ancestors**: let $\mathrm{An}(X)$ be causal ancestors of $X$, i.e., for every $X_j \in \mathrm{An}(X_i)$, we write $X_j \dashrightarrow X_i$.
- **Descendants**: let $\mathrm{De}(X)$ be causal descendants of $X$, i.e., for every $X_j \in \mathrm{De}(X_i)$, we write $X_i \dashrightarrow X_j$.

In addition, nodes can also be classified in terms of triadic relationships.

- $X_i$ is a **confounder** of $X_j, X_k$ if and only if $X_i \in \mathrm{Pa}(X_j) \cap \mathrm{Pa}(X_k)$. This corresponds to the causal path structure $X_j \leftarrow X_i \to X_k$, which is called a **fork structure**.
- $X_i$ is a **mediator** of $X_j, X_k$ if and only if $X_i \in \mathrm{Ch}(X_j) \cap \mathrm{Pa}(X_k)$. This corresponds to the causal path structure $X_j \to X_i \to X_k$, which is called a **chain structure**.
- $X_i$ is a **collider** of $X_j, X_k$ if and only if $X_i \in \mathrm{Ch}(X_j) \cap \mathrm{Ch}(X_k)$. This corresponds to the causal path structure $X_j \to X_i \leftarrow X_k$, which is called a **v-structure**.

Any causal graph is composed of the three above structures. Based on these structures, we also introduce the idea of open and blocked paths between two nodes to discuss causal relationships. Generally, these paths are described by the d-separation criterion.

**Definition 1** (d-separation Pearl (1988)). *A path $U$ in a DAG $\mathcal{G}$ is said to be **blocked** by a set of nodes* **S** *if either,*

- *$U$ contains a fork structure $X_j \leftarrow X_i \to X_k$ or a chain structure $X_j \to X_i \to X_k$ such that the middle node $X_i$ is in* **S***,*
- *$U$ contains a v-structure $X_j \to X_i \leftarrow X_k$ such that the collider node $X_i$ is not in* **S***, and neither is any descendant $X_l \in \mathrm{De}(X_i)$.*

*If* **S** *blocks every path between two nodes, then they are d-separated given* **S***, and thus are independent conditional on* **S***. Such a relation is denoted by $X_j \bowtie X_k \,|\, \mathbf{S}$.*

Conversely, a path which is not blocked is called an **open** path, and when there is at least one open path between two nodes, they are not d-separated, i.e., some information is shared between the two. Also, we need to estimate the above relations from multivariate time series data. We use $\perp\!\!\!\perp$ for an independent relation in a dataset.

## A.2  MISSINGNESS MECHANISMS

Missing values $\boldsymbol{V}^m$ in observational data do not always occur at random. Missingness mechanisms can be commonly categorized into the following three classes according to the factors behind the missing data.

- Data are **Missing Completely At Random (MCAR)** if the reasons for missing values $\boldsymbol{V}^m$ are independent of any variables, i.e., $\{\boldsymbol{V}^o, \boldsymbol{V}^m\} \bowtie \mathbf{R}$.
- Data are **Missing At Random (MAR)** if the reasons for missing values $\boldsymbol{V}^m$ depend only on the observed data $\boldsymbol{V}^o$ and not on the missing values themselves, i.e., $\boldsymbol{V}^m \bowtie \mathbf{R} \,|\, \boldsymbol{V}^o$. For example, women are relatively more likely to omit their age.
- Data are **Missing Not At Random (MNAR)** if it is neither MAR nor MCAR. This is also referred to as non-ignorable missing.

## B  RELATED WORK

In this section, we briefly describe investigations related to our work. As mentioned in the main text, Table 1 summarizes six relative advantages of CANMI.

Causal discovery from observational data has attracted huge interest for many applications including recommendation systems Wang et al. (2020); Gao et al. (2024), epidemiology Robins et al. (2000), and others Kyono et al. (2021); Chihara et al. (2025). In general, typical causal discovery approaches fall mainly into three categories: constraint-based methods Spirtes et al. (1995); Colombo & Maathuis (2014), score-based methods Zheng et al. (2019); Li et al. (2024), and functional causal model (FCM)-based methods Peters et al. (2014). Constraint-based methods, such as PC-algorithm Spirtes et al. (1993), use CI tests to identify a causal structure by removing irrelevant edges and orienting the remaining edges based on separation rules. Score-based methods assess causal structure candidates by assigning them scores based on predefined scoring criteria (e.g., BIC Schwarz (1978)), and select candidates that achieve the highest scores as the most plausible results. GES Chickering (2002) is a traditional score-based Bayesian algorithm that discovers causal relationships in a greedy manner, and Generalized-GES (GGES) Huang et al. (2018) extends it to nonlinear settings via kernel-based score functions. NOTEARS Zheng et al. (2018) introduces a differentiable optimization method for discovering directed acyclic graphs by substituting the acyclic combinatorial constraint with a continuous regularization formulation. FCM-based methods take a different approach by assuming specific functional forms for the data-generating mechanisms, such as additive noise models (ANMs) Hoyer et al. (2008) or post-nonlinear (PNL) models Zhang & Chan (2006); Zhang & Hyvarinen (2009). LiNGAM Shimizu et al. (2006) assumes a linear acyclic model with non-Gaussian errors and utilizes this assumption to identify the causal structure. However, most of these approaches rely on assumptions of time independence and completeness in the observed data.

Alternative approaches have been developed to address temporal dependencies more explicitly Pamfil et al. (2020); Hasan et al. (2023); Wu et al. (2024); Chihara et al. (2025). Granger causality Granger (1969) is one of the classical statistical approaches for causality in time series data and has been widely utilized over decades Wei et al. (2023), but Granger causality only indicates the presence of a predictive relationship Granger & Newbold (1986); Peters et al. (2017), so it is then different from true causality. PCMCI Runge et al. (2019) and its derivatives Gerhardus & Runge (2020); Runge (2020); Ferdous et al. (2023); Gao et al. (2023) handle autocorrelation, which is a major source of false positives in time series causal discovery. However, most of the above approaches make a stationary assumption. PCMCI$_\Omega$ Gao et al. (2023) is the first causal discovery algorithm that is capable of handling semi-stationary time series with periodically recurring causal mechanisms. However, it assumes that structural changes follow a fixed periodic pattern, which may limit its applicability in practice. CD-NOD Huang et al. (2020) is a framework designed to discover causal relations from heterogeneous or nonstationary data by exploiting distribution shifts across domains or over time. SpaceTime Mameche et al. (2025) discovers distinct temporal regimes and context-specific causal structures from nonstationary multivariate time series using the score built on the minimum description length (MDL) principle Rissanen (1978), and Gaussian processes for modeling causal relationships. However, they implicitly assume that nonstationary time series data consist of multiple piecewise-stationary segments, then cannot handle continuous change over time effectively.

Real-world datasets generally contain missing values for various reasons, and it is necessary to develop causal discovery approaches from partially observed data. MissDAG Gao et al. (2022) utilizes an EM-based paradigm for causal discovery in the presence of missing values and models the simpler noise distributions instead of directly modeling the complex likelihood of the partially observed samples with the additive noise models, but it only focuses on MCAR data. Recent studies have increasingly stressed that missing values often reflect systematic patterns driven by an underlying causal structure, rather than occurring purely at random Mohan et al. (2013); Ma & Chen (2019); Ma & Zhang (2021). MVPC Tu et al. (2019) focuses on the task of causal discovery from observations with non-random missing values. However, it implicitly assumes that the data distribution and missingness mechanisms are time-independent.

To the best of our knowledge, this is the first work to propose an algorithm specifically designed for causal discovery from partially observed time series data with nonstationary missingness mechanisms and establish theoretical guarantees including soundness and recoverability of the joint distribution.

## C    PRELIMINARIES

### C.1    INTUITIVE DESCRIPTIONS OF ASSUMPTIONS

We describe why the assumptions provided in Section 2.3 are required and clarify that these assumptions are minimal for identifiability.

#### C.1.1    ASSUMPTION 1

Assumption 1 states that all unobserved confounders responsible for nonstationarity can be represented as smooth functions of the time index. This formulation effectively captures practically common phenomena where dominant changes are driven by low-dimensional, smoothly time-varying factors, such as seasonal effects and user behavior trends. This smoothness constraint allows us to model complex temporal patterns, including nonlinear and periodic behaviors, without relying on a restrictive parametric form. Furthermore, Assumption 1 rules out scenarios in which latent factors cannot be expressed as smooth functions of the time index, such as abrupt changes or instantaneous external shocks.

#### C.1.2    ASSUMPTION 5

Assumption 5 states that missingness neither obscures true CI relations nor introduces spurious independencies. This assumption rules out two pathological cases: (i) spurious independences created purely by selection through missingness and (ii) true independences being masked in the observed marginal. Pathology would mislead constraint-based methods (e.g., either by erroneously deleting a true edge or by retaining a spurious one).

Assumption 5 can be viewed as a natural extension of the classical faithfulness principle (i.e., Assumption 3) to settings involving non-random missing data, thereby placing it in alignment with established frameworks in causal discovery. Several previous studies have adopted this assumption Tu et al. (2019); Dai et al. (2024); Strobl et al. (2018). In particular, Dai et al. (2024) said this assumption is rarely violated. Thus, rather than being overly strong, Assumption 5 can be viewed as a necessary and practically satisfied condition for ensuring identifiability in realistic settings.

#### C.1.3    ASSUMPTION 7

Assumption 7 states that the probability of missingness for a variable $X$ is not affected by the value of $X$. In well-designed data collection, whether $X$ is observed is decided a priori by protocol and before observing its value itself. Assumption 7 formalizes this practice and removes a rare but anomalous case that would otherwise entangle the missingness mechanism with latent values in a way that undermines identifiability, so this assumption reflects standard measurement practice rather than an ad-hoc modeling choice. In fact, some existing methods also utilize the no-self masking missingness assumption Tu et al. (2019); Kyono et al. (2021). For example, we consider clinical data, such as *(blood pressure, Duke Treadmill Score (DTS), age)*. To mitigate potential harm, physicians may cancel physically demanding examinations, such as a treadmill exercise test (TET), for hypertensive or older patients. Since the DTS is obtained with the TET, the presence or absence of the DTS is determined by blood pressure and age, rather than the score itself.

In addition, note that when a missingness indicator $R_X$ is influenced solely by $X$, our algorithm returns the true causal relationship. For example, we consider a causal graph where $\boldsymbol{V} = \{X, Y\}$ and only the edge $X \rightarrow R_X$ exists. In this case, under Assumptions 3 and 5, it is clear that $X \perp\!\!\!\perp Y \Leftrightarrow X \perp\!\!\!\perp Y \mid R_X = 0$ holds. Without Assumption 7, it is challenging to identify the causal relationship where $X \rightarrow R_X$ exists and other variables also have an influence on $R_X$. Thus, although Assumption 7 rules out only a narrow class of missingness mechanisms, specifically self-masking, our algorithm remains widely applicable across real-world scenarios.

## D    PROOFS

**Proposition 1.** *Suppose that for $X, Y \in \boldsymbol{V}$, $\mathbf{S} \subseteq \boldsymbol{V}$, and latent factors $\boldsymbol{U}$ that drive the nonstationary changes, even if it holds that $X \perp\!\!\!\perp Y \mid \mathbf{S} \cup \boldsymbol{U}$, it does not necessarily hold $X \perp\!\!\!\perp Y \mid \mathbf{S} \cup \{R_X = 0, R_Y = 0, \boldsymbol{R_S} = \boldsymbol{0}\}$.*

*Proof.* We prove this by constructing a counterexample. Consider a causal graph, where $U$ is a common latent factor of $X$ and $R_X$, and $\mathbf{S}$ is an empty set, i.e., $X \leftarrow U \rightarrow R_X$. In addition, let $Y$ be a parent of $R_X$. In this graph, the path between $X$ and $Y$ is written as $X \leftarrow U \rightarrow R_X \leftarrow Y$, indicating that $X \perp\!\!\!\perp Y \mid U$ according to Definition 1. Here, consider the conditional independence given the missingness indicators $\{R_X = 0, R_Y = 0, \boldsymbol{R_S} = \boldsymbol{0}\}$. In this case, the set of conditioning variables includes the collider $R_X$. Conditioning on a collider $R_X$ unblocks the v-structure $U \rightarrow R_X \leftarrow Y$ by Definition 1. Since the latent factor $U$ is unobserved, the path $X \leftarrow U \rightarrow R_X \leftarrow Y$ becomes open. This induces a spurious dependency between $X$ and $Y$. Therefore, although $X \perp\!\!\!\perp Y \mid U$ holds, it does not necessarily hold $X \perp\!\!\!\perp Y \mid \mathbf{S} \cup \{R_X = 0, R_Y = 0, \boldsymbol{R_S} = \boldsymbol{0}\}$. $\qquad\square$

**Proposition 2.** *Under Assumptions 1-7, the direct causes of missingness indicators are identifiable.*

*Proof.* Under Assumption 7, it suffices to consider the case where we can test whether a variable $X_i \in \boldsymbol{V}$ is a direct cause of the missingness indicator $R_j$ that $i \neq j$. First, when there is a dependency $X_i \rightarrow R_j$, then by definition, this relationship holds regardless of any external influence and can be detected through CI tests. In contrast, when there is no direct edge between $X_i$ and $R_j$, we need to identify the CI relation between the two nodes, but a spurious dependency may occur due to unobserved confounders of nonstationary causal mechanisms, and shared confounders. With Assumption 1, such an influence can be written as smooth functions of time $\{u(T)\}$, where each $u$ is a function of the time index $T$. These functions can be thought of as covering all possible confounders. Then, there exists $\mathbf{S} \subseteq \boldsymbol{V} \setminus \{X_i, X_j\}$ such that the true CI relation can be described as

$$X_i \perp\!\!\!\perp R_j \mid \mathbf{S} \cup \{u(T)\}. \tag{4}$$

Since $T$ is a surrogate variable and it is not part of a causal system, $T$ is also not a descendant of $X_i$, so it holds by Assumptions 2 and 3,

$$X_i \perp\!\!\!\perp T \mid \mathbf{S} \cup \{u(T)\}. \tag{5}$$

Both Eq. (4) and Eq. (5) indicate

$$X_i \perp\!\!\!\perp R_j \mid \mathbf{S} \cup \{u(T)\} \cup \{T\}. \tag{6}$$

Here, since each $u(T)$ is a deterministic function, we have $\sigma(u(T)) \subseteq \sigma(T)$, meaning that the information carried by $u(T)$ is measurable with respect to the $\sigma$-algebra generated by $T$. Eq. (6) is then equivalent to $X_i \perp\!\!\!\perp R_j \mid \mathbf{S} \cup \{T\}$. Therefore, the direct causes of missingness indicators are identifiable under nonstationary missingness mechanisms. $\qquad\square$

**Proposition 3.** *Under Assumptions 1-7, for any $X, Y \in \boldsymbol{V}$, and $\mathbf{S} \subseteq \boldsymbol{V} \cup \{T\} \setminus \{X, Y\}$, if it holds that $X \perp\!\!\!\perp Y \mid \mathbf{S} \cup \{R_X = 0, R_Y = 0, \boldsymbol{R_S} = \boldsymbol{0}\}$, it also holds that $X \perp\!\!\!\perp Y \mid \mathbf{S}$, where $T$ is the time index and $\boldsymbol{R_S} = \boldsymbol{0}$ means that every $R \in \boldsymbol{R_S}$ takes the value zero.*

*Proof.* In this proof, we derive that the following conditional independence implication holds

$$X \perp\!\!\!\perp Y \mid \mathbf{S} \cup \{R_X = 0, R_Y = 0, \boldsymbol{R_S} = \boldsymbol{0}\} \implies X \perp\!\!\!\perp Y \mid \mathbf{S}. \tag{7}$$

First, with Assumption 5, the above condition is equivalent to $X \perp\!\!\!\perp Y \mid \mathbf{S} \cup \{R_X, R_Y, \boldsymbol{R_S}\}$. And, with Assumptions 4 and 6, every missingness indicator $R$ can only be a leaf node in an m-graph, in other words, $R$ becomes neither a confounder nor a mediator for any pair of nodes. Note that if $R$ becomes a collider of $X, Y$, then $R$ cannot open any path because removing variables from a conditioning set $\mathbf{S}$ does not open paths containing a v-structure. Thus, if $X$ and $Y$ are conditionally independent given the missingness indicators, then this CI relation still holds in spite of the absence of the missingness indicators in the conditioning set according to Definition 1. Hence, it holds that $X \perp\!\!\!\perp Y \mid \mathbf{S}$. $\qquad\square$

**Proposition 4.** *Suppose that $X$ and $Y$ are not adjacent in a true m-graph, and that for any $\mathbf{S} \subseteq \boldsymbol{V} \cup \{T\} \setminus \{X, Y\}$ such that $X \perp\!\!\!\perp Y \mid \mathbf{S}$, it holds $X \not\perp\!\!\!\perp Y \mid \mathbf{S} \cup \{R_X = 0, R_Y = 0, \boldsymbol{R_S} = \boldsymbol{0}\}$. Then, under Assumptions 1-7, for at least one variable $Z \in \{X\} \cup \{Y\} \cup \mathbf{S}$, the missingness indicator $R_Z$ is either the direct common effect or a descendant of the direct common effect of $X$ and $Y$.*

*Proof.* We separate the proof into Necessity and Sufficiency. Let $U_{XY}$ be an arbitrary path between $X$ and $Y$, and $W$ be a collider on the path $U_{XY}$.

*Necessary*: Suppose that $\forall \mathbf{S} \subseteq \mathbf{V} \cup \{T\} \setminus \{X, Y\} : X \perp\!\!\!\perp Y \,|\, \mathbf{S} \Rightarrow X \not\perp\!\!\!\perp Y \,|\, \mathbf{S} \cup \{R_X = 0, R_Y = 0, \mathbf{R_S} = \mathbf{0}\}$ holds. First, with Assumption 5, it holds that $X \not\perp\!\!\!\perp Y \,|\, \mathbf{S} \cup \{R_X = 0, R_Y = 0, \mathbf{R_S} = \mathbf{0}\} \Leftrightarrow X \not\perp\!\!\!\perp Y \,|\, \mathbf{S} \cup \{R_X, R_Y, \mathbf{R_S}\}$. Here, since it holds that $X \perp\!\!\!\perp Y \,|\, \mathbf{S}$, as shown in Definition 1, every path between $X$ and $Y$ is blocked by the d-separation definition, either (i) at least one of the confounders or mediators on the path $U_{XY}$ lies in $\mathbf{S}$, (ii) a collider node $W$ on the path $U_{XY}$ and all its descendants are not in $\mathbf{S}$. Since we consider that conditioning additionally on the missingness indicators $\{R_X, R_Y, \mathbf{R_S}\}$ opens a previously blocked path between $X$ and $Y$, it is sufficient to only focus on the second condition of the d-separation rules. Hence, for the path $U_{XY}$ to open, there must exist $Z \in \{X\} \cup \{Y\} \cup \mathbf{S}$ such that $R_Z$ is either the collider $W$ itself or a descendant of $W$. In addition, under Assumptions 4 and 6, since any missing indicator is only affected by variables in $\mathbf{V} \cup \{T\}$ and cannot affect others, we claim that a collider $W$ on the path $U_{XY}$ must be a direct common effect with structure $X \to W \leftarrow Y$; otherwise one could choose $\mathbf{S}^\star$ to block the path $U_{XY}$ even when conditioning on the missingness indicators $\{R_X, R_Y, \mathbf{R_S}\}$. Specifically, if a collider $W$ is not a direct common effect, then there exists a first non-collider node $B$ that appears when starting from $X$ or $Y$ toward $W$ along with the path $U_{XY}$. Choosing $\mathbf{S}^\star$ including the node $B$ yields $X \perp\!\!\!\perp Y \,|\, \mathbf{S}^\star$ by Definition 1. And, this CI relation is preserved even after the addition of $\{R_X, R_Y, \mathbf{R_S}\}$. Therefore, for at least one variable $Z \in \{X\} \cup \{Y\} \cup \mathbf{S}$, the missingness indicator $R_Z$ is either the direct common effect or a descendant of the direct common effect of $X$ and $Y$.

*Sufficiency*: Conversely, suppose there exists $Z \in \{X\} \cup \{Y\} \cup \mathbf{S}$ whose missingness indicator $R_Z$ is either the direct common effect or a descendant of the direct common effect of $X$ and $Y$; then there exists a direct common effect $W$, it holds that $R_Z \in \mathrm{De}(W)$ or $R_Z = W$, and there exists a path $U_{XY}$ which has the structure written as $X \to W \leftarrow Y$. Since $X \perp\!\!\!\perp Y \,|\, \mathbf{S}$ holds, neither $W$ nor any of its descendants are in $\mathbf{S}$, and the path $U_{XY}$ is blocked at $W$. When we additionally condition on $\{R_X, R_Y, \mathbf{R_S}\}$, the missingness indicator $R_Z$ is included in the conditioning set $\mathbf{S} \cup \{R_X, R_Y, \mathbf{R_S}\}$. Since $R_Z = W$ or $R_Z \in \mathrm{De}(W)$, conditioning $R_Z$ opens the $X \to W \leftarrow Y$ path, implying

$$X \not\perp\!\!\!\perp Y \,|\, \mathbf{S} \cup \{R_X = 0, R_Y = 0, \mathbf{R_S} = \mathbf{0}\},$$

under Assumption 5. Moreover, by Assumption 4, missingness indicators have no children, so adding them as leaf nodes cannot spuriously open non-collider paths. This ensures that the collider at $W$ only opens the path $U_{XY}$.

In summary, we conclude that when conditioning additionally on the missingness indicators $\{R_X = 0, R_Y = 0, \mathbf{R_S} = \mathbf{0}\}$ opens a path $U_{XY}$ between the nodes $X$ and $Y$, the missingness indicator $R_Z$ is either the direct common effect or a descendant of the direct common effect of $X$ and $Y$. $\qquad\square$

**Theorem 1.** *Under Assumptions 1-7, given the parents of each missingness indicator* $\mathrm{Pa}(R_i)$*, the joint distribution* $P(\mathbf{V})$ *is recoverable, and we then have*

$$P(\mathbf{V}) = \frac{P(\mathbf{R} = \mathbf{0}, \mathbf{V})}{\prod_i P(R_i = 0 \,|\, \mathrm{Pa}^+(R_i), \mathbf{R}_{\mathrm{Pa}(R_i)} = 0)}$$

$$= \frac{1}{Z} P(\mathbf{V} \,|\, \mathbf{R} = \mathbf{0}) \prod_i \omega_{\mathrm{Pa}(R_i)}$$

*where*

$$\mathrm{Pa}^+(R_i) = \mathrm{Pa}(R_i) \cup \{T\} \qquad \textit{(time-augmented parents)},$$

$$Z = \frac{\prod_i P(R_i = 0 \,|\, \mathbf{R}_{\mathrm{Pa}(R_i)} = \mathbf{0})}{P(\mathbf{R} = \mathbf{0})} \qquad \textit{(normalizing constant)},$$

$$\omega_{\mathrm{Pa}(R_i)} = \frac{P(\mathrm{Pa}^+(R_i) \,|\, \mathbf{R}_{\mathrm{Pa}(R_i)} = \mathbf{0})}{P(\mathrm{Pa}^+(R_i) \,|\, R_i = 0, \mathbf{R}_{\mathrm{Pa}(R_i)} = \mathbf{0})} \qquad \textit{(density ratio weights)}.$$

*Proof.* The observed joint distribution $P(\boldsymbol{V} \mid \boldsymbol{R} = \boldsymbol{0})$ can be described according to the graph $\mathcal{G}$ as

$$P(\boldsymbol{V} \mid \boldsymbol{R} = \boldsymbol{0}) = \frac{P(\boldsymbol{R} = \boldsymbol{0} \mid \boldsymbol{V})P(\boldsymbol{V})}{P(\boldsymbol{R} = \boldsymbol{0})}$$

$$P(\boldsymbol{V}, \boldsymbol{R} = \boldsymbol{0}) = \sum_{\boldsymbol{U}} P(\boldsymbol{V}, \boldsymbol{U})P(\boldsymbol{R} = \boldsymbol{0} \mid \boldsymbol{V}, \boldsymbol{U})$$

$$P(\boldsymbol{V}, \boldsymbol{R} = \boldsymbol{0}) = P(\boldsymbol{V}) \sum_{\boldsymbol{U}} P(\boldsymbol{U} \mid \boldsymbol{V})P(\boldsymbol{R} = \boldsymbol{0} \mid \boldsymbol{V}, \boldsymbol{U}).$$

Under Assumption 6, the above formula is equivalent to

$$P(\boldsymbol{V}, \boldsymbol{R} = \boldsymbol{0}) = P(\boldsymbol{V}) \sum_{\boldsymbol{U}} P(\boldsymbol{U} \mid \boldsymbol{V}) \left\{ \prod_i P(R_i = 0 \mid \mathrm{Pa}(R_i), \mathrm{Pa}^m(R_i)) \right\}.$$

With Assumption 1, we only have the unobserved confounders written as smooth functions of time. Accordingly, we can replace $\mathrm{Pa}^m(R_i)$ with $\mathbf{u}_i(T)$, which is a deterministic function of $T$. Note that in the case of no unobserved confounders for $R_i$, $\mathbf{u}_i(T)$ is an empty set (i.e., $\mathbf{u}_i(T) = \emptyset$). Under Assumptions 2 and 3 it follows that $R_i \perp\!\!\!\perp T \mid \mathrm{Pa}(R_i) \cup \mathbf{u}_i(T)$. Consequently, the conditional probability satisfies

$$P(R_i = 0 \mid \mathrm{Pa}(R_i), \mathbf{u}_i(T)) = P(R_i = 0 \mid \mathrm{Pa}(R_i), \mathbf{u}_i(T), T).$$

Here, since each component $u(T) \in \mathbf{u}_i(T)$ is a deterministic function of $T$, it holds $\sigma(\mathbf{u}_i(T)) \subseteq \sigma(T)$, meaning that the information carried by $u(T)$ is measurable with respect to the $\sigma$-algebra generated by $T$. Hence, it holds that

$$P(R_i = 0 \mid \mathrm{Pa}(R_i), \mathbf{u}_i(T), T) = P(R_i = 0 \mid \underbrace{\mathrm{Pa}(R_i), T}_{\mathrm{Pa}^+(R_i)}),$$

where $\mathrm{Pa}^+(R_i) = \mathrm{Pa}(R_i) \cup \{T\}$. Based on the above, we have that the joint distribution can be decomposed as

$$P(\boldsymbol{V}, \boldsymbol{R} = \boldsymbol{0}) = P(\boldsymbol{V}) \sum_{\boldsymbol{U}} P(\boldsymbol{U} \mid \boldsymbol{V}) \left\{ \prod_i P(R_i = 0 \mid \mathrm{Pa}^+(R_i)) \right\}$$

$$= P(\boldsymbol{V}) \prod_i P(R_i = 0 \mid \mathrm{Pa}^+(R_i)),$$

where $\sum_{\boldsymbol{U}} P(\boldsymbol{U} \mid \boldsymbol{V}) = 1$ because $\prod_i P(R_i = 0 \mid \mathrm{Pa}^+(R_i))$ dose not depend on $\boldsymbol{U}$. Under Assumption 7, i.e., $X_i$ is not a parent of $R_i$, we have $R_i \perp\!\!\!\perp \boldsymbol{R}_{\mathrm{Pa}(R_i)} \mid \mathrm{Pa}(R_i) \cup T$, therefore,

$$P(\boldsymbol{V}, \boldsymbol{R} = \boldsymbol{0}) = P(\boldsymbol{V}) \prod_i P(R_i = 0 \mid \mathrm{Pa}^+(R_i), \boldsymbol{R}_{\mathrm{Pa}(R_i)} = \boldsymbol{0}).$$

Solving for $P(\boldsymbol{V})$ from the equality above and expanding the terms via Bayes rule, we obtain

$$P(\boldsymbol{V}) = \frac{P(\boldsymbol{V}, \boldsymbol{R} = \boldsymbol{0})}{\prod_i P(R_i = 0 \mid \mathrm{Pa}^+(R_i), \boldsymbol{R}_{\mathrm{Pa}(R_i)} = \boldsymbol{0})}$$

$$= \frac{P(\boldsymbol{V} \mid \boldsymbol{R} = \boldsymbol{0})P(\boldsymbol{R} = \boldsymbol{0})}{\prod_i \dfrac{P(R_i = 0, \mathrm{Pa}^+(R_i) \mid \boldsymbol{R}_{\mathrm{Pa}(R_i)} = \boldsymbol{0})}{P(\mathrm{Pa}^+(R_i) \mid \boldsymbol{R}_{\mathrm{Pa}(R_i)} = \boldsymbol{0})}}$$

$$= \frac{P(\boldsymbol{V} \mid \boldsymbol{R} = \boldsymbol{0})P(\boldsymbol{R} = \boldsymbol{0})}{\prod_i \dfrac{P(\mathrm{Pa}^+(R_i) \mid R_i = 0, \boldsymbol{R}_{\mathrm{Pa}(R_i)} = \boldsymbol{0})P(R_i = 0 \mid \boldsymbol{R}_{\mathrm{Pa}(R_i)} = \boldsymbol{0})}{P(\mathrm{Pa}^+(R_i) \mid \boldsymbol{R}_{\mathrm{Pa}(R_i)} = \boldsymbol{0})}}$$

$$= P(\boldsymbol{V} \mid \boldsymbol{R} = \boldsymbol{0}) \cdot \underbrace{\frac{P(\boldsymbol{R} = \boldsymbol{0})}{\prod_i P(R_i = 0 \mid \boldsymbol{R}_{\mathrm{Pa}(R_i)} = \boldsymbol{0})}}_{\text{normalizing constant } Z} \cdot \underbrace{\prod_i \frac{P(\mathrm{Pa}^+(R_i) \mid \boldsymbol{R}_{\mathrm{Pa}(R_i)} = \boldsymbol{0})}{P(\mathrm{Pa}^+(R_i) \mid R_i = 0, \boldsymbol{R}_{\mathrm{Pa}(R_i)} = \boldsymbol{0})}}_{\text{density ratio } \omega_{\mathrm{Pa}(R_i)}}.$$

Hence, the desired result is obtained. $\qquad\square$

**Theorem 2.** *Under Assumptions 1-7, the skeleton result from the CANMI algorithm is independent of the order of variables* $(X_1, \ldots, X_d)$.

*Proof.* This proof is straightforward and mainly follows the theoretical result found in Colombo & Maathuis (2014). CI tests in our algorithm are conducted in stages according to the cardinality $\ell = |\mathbf{S}|$ of the conditioning sets. Specifically, for each $\ell$, conditioning sets are formed from adjacency sets that are kept fixed throughout the loop $|\mathbf{S}| = \ell$, and no edge is deleted within the loop. Since the family of CI tests considered at the loop $|\mathbf{S}| = \ell$ does not depend on the order of variables in which they are executed, the set of edges marked for deletion depends only on the CI test results, not on their execution order; hence, the post-stage skeleton is uniquely determined. By iterating this procedure over $\ell$, the final skeleton is uniquely determined as well. Hence, our algorithm is order-independent. $\square$

**Theorem 3.** *Under Assumptions 1-7, CANMI returns a causal skeleton graph that is exactly consistent with the true causal skeleton.*

*Proof.* First, for each pair $X, Y \in \mathbf{V}$, our proposed algorithm deletes an edge between $X$ and $Y$ if there is a subset $\mathbf{S} \subseteq \mathbf{V} \cup \{T\}$ such that $X \perp\!\!\!\perp Y \mid \mathbf{S}$ in Step 3. Here, the algorithm determines whether the time index $T$ is included in the conditioning set $\mathbf{S}$ according to the outputs of Step 1. According to Proposition 3, while extraneous edges may occur, an edge is deleted in Step 3 only if it is absent in the true graph. Also, according to Proposition 4, it identifies all extraneous edge candidates in the estimated graph obtained after Step 3. Next, in Step 4, for each such candidate edge, it recovers the data distribution using the recoverability formula in Theorem 1, and performs CI tests on a modified dataset obtained by importance resampling drawn from the reconstructed distribution. By Assumption 3, CI relations inferred from the reconstructed data imply true d-separations in the true causal graph. Therefore, it eliminates all false positives and preserves all true causal dependencies. Hence, the estimated skeleton graph returned by CANMI is exactly consistent with the true causal skeleton under Assumptions 1-7. $\square$

# E  EXPERIMENTS

## E.1  EXPERIMENTAL SETUP

**Computing infrastructure.** We conducted all our experiments on a system running Ubuntu 22.04.4 LTS (GNU/Linux 5.15.0-97-generic x86_64), equipped with $2 * $ Intel Xeon Platinum 8268 2.7GHz 24-core CPUs, $8 * 64$GB DDR4 RAM, and $2 * $ NVIDIA RTX A6000 GPUs.

**Baselines.** We compared the following eight methods for causal discovery.

- SpaceTime Mameche et al. (2025): discovers causal relationships from nonstationary time series data by detecting multiple piecewise-stationary segments.

- MissDAG Gao et al. (2022): is a general EM-based framework in the presence of missing data, which leverages identifiable additive noise models (ANMs) and penalized likelihood optimization.

- CD-NOD Huang et al. (2020): is a nonparametric approach designed to recover the skeleton and determine orientations of the causal structure over nonstationary observed variables.

- MVPC Tu et al. (2019): is a correction-based extension of the PC-algorithm, which removes extraneous edges generated due to missingness mechanisms underlying the causal process.

- NOTEARS-MLP (NO-MLP, for short) Zheng et al. (2019): is an extension of NOTEARS Zheng et al. (2018) (mentioned below) for nonlinear settings, which aims to approximate the generative structural equation model by MLP.

- NOTEARS Zheng et al. (2018): is a differentiable optimization method with an acyclic regularization term to estimate the structure of a directed acyclic graph.

- GGES Huang et al. (2018): provides generalized score functions for causal discovery based on the characterization of general CI relations without assuming specific model classes.

- LiNGAM Shimizu et al. (2006): is a traditional FCM-based causal discovery approach with a linear non-Gaussian acyclic model.

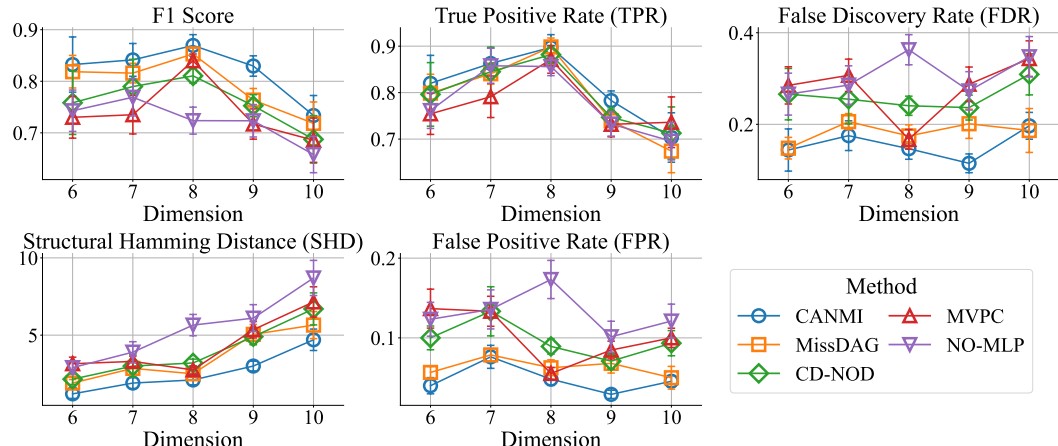

Figure 2: Causal discovery results for synthetic datasets with various $d \in \{6, 7, 8, 9, 10\}$.

**Implementation details.** For the constraint-based methods, we used the KCI-test Zhang et al. (2011) for CI tests and we set the significance level at $\alpha = 0.05$. For MissDAG, we reported the results for the Non-Linear (NL)-ANM case. And we utilized Zero-order Holder to evaluate approaches, which cannot handle missing values in observed data. Finally, we conducted all our experiments with five different seeds for a fair comparison.

Furthermore, regarding the baselines used in this paper, we mainly employed the publicly available implementations of the baseline methods to ensure reproducibility and consistency with prior work. The implementations of SpaceTime[2], MissDAG[3], NOTEARS[4], NO-MLP[4], and LiNGAM[5] were obtained from the authors' original repositories. We utilized the implementations of CD-NOD, MVPC, and GGES from the causal-learn package Zheng et al. (2023), which is available at https://github.com/py-why/causal-learn.

### E.2 ROBUSTNESS

**Setup.** We conducted the experiments on various numbers of variables and missing probabilities. Specifically, we varied the dimensionality $d \in \{6, 7, 8, 9, 10\}$ and the missing probability $p_h \in \{0.0, 0.4, 0.6, 0.8\}$ while fixing the sample size $N = 3000$ and the average degree $k_d = 2$.

**Results.** Figure 2 shows results for synthetic datasets with dimensionalities $d \in \{6, 7, 8, 9, 10\}$ for CANMI and competing methods. Overall, our proposed method outperforms the competitive baselines across all evaluation metrics. We can also see that CANMI partly degrades for high-dimensional data, as mentioned in the limitations. Nevertheless, some baselines already fail to discover causal relationships in low dimensions, while others experience a similar decline in discovery accuracy as dimensionality increases. Thus, CANMI is relatively robust compared with its baselines against various dimensionalities.

### E.3 STATISTICAL ANALYSIS

Figure 3 shows critical difference diagrams for F1 Score and SHD. These diagrams are based on the Wilcoxon-Holm method Wilcoxon (1945), where methods not connected by a bold line are sufficiently different regarding their average rank. We can observe that the significant improvements that CANMI achieved over its baselines are valid according to statistical tests.

---

[2]https://github.com/srhmm/spacetime

[3]https://github.com/ErdunGAO/MissDAG

[4]https://github.com/xunzheng/notears

[5]https://github.com/cdt15/lingam

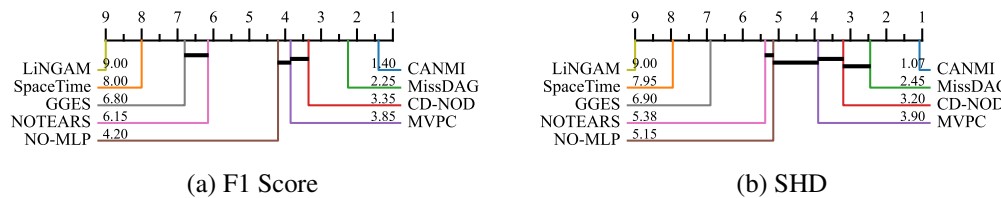

(a) F1 Score                    (b) SHD

Figure 3: Critical difference diagrams of causal discovery results. The ticks at the top indicate the average rank of each method (smaller is better).

### E.4 SENSITIVITY ANALYSIS

We conducted a sensitivity analysis with respect to the kernel functions and bandwidth selection rules of kernel density estimation (KDE) to compute the density ratio weights in Theorem 1. We considered four kernel functions (Gaussian, Epanechnikov, Tophat, and Exponential) and two bandwidth selection rules (Scott's Rule and Silverman's Rule). Table 4 reports the results across different kernels and bandwidth rules. We can observe that the performance of CANMI remains consistently high performance, with negligible variance across all evaluation metrics. This indicates that our proposed method is robust with respect to moderate changes in KDE hyperparameters.

Table 4: Sensitivity analysis results.

| Metric | | TPR ($\uparrow$) | FPR ($\downarrow$) | FDR ($\downarrow$) | F1 ($\uparrow$) | SHD ($\downarrow$) |
|---|---|---|---|---|---|---|
| Epanechnikov | Scott | 0.774 | 0.0452 | 0.0936 | 0.828 | 4.40 |
| | Silverman | 0.770 | 0.0452 | 0.0936 | 0.826 | 4.47 |
| Exponential | Scott | 0.776 | 0.0429 | 0.0917 | 0.828 | 4.40 |
| | Silverman | 0.776 | 0.0429 | 0.0917 | 0.828 | 4.40 |
| Gaussian | Scott | 0.774 | 0.0452 | 0.0929 | 0.828 | 4.40 |
| | Silverman | 0.770 | 0.0452 | 0.0936 | 0.826 | 4.47 |
| Tophat | Scott | 0.777 | 0.0452 | 0.0929 | 0.828 | 4.40 |
| | Silverman | 0.777 | 0.0452 | 0.0936 | 0.828 | 4.40 |

### E.5 COMPUTATIONAL EFFICIENCY

We measured wall-clock time and peak memory usage for synthetic datasets of length $N = 1,000$ and varying dimensions $d \in \{5, 10, 20, 50, 100\}$. Figure 4 shows the computational efficiency of CANMI and its competitive baselines (i.e., MissDAG and CD-NOD). Although, as mentioned in the limitation, the worst-case time complexity of CANMI is exponential in theory, our empirical experiments demonstrated that the computational time needed for our method is competitive with its baselines, and its memory usage grows only moderately with increasing dimensionality. This slight increase primarily stems from the use of kernel density estimation in the density ratio adjustment step. First, unlike MissDAG, which relies on iterative EM procedures over latent distributions, CANMI adopts a constraint-based framework that avoids costly likelihood-based optimization, reducing memory overhead. Second, compared with CD-NOD, which does not account for missing values, spurious edges caused by missingness remain unpruned, leading to an increased number of CI tests and larger conditioning sets.

## F LIMITATIONS

The proposed method has some limitations that may be interesting to address in future work. First, as with other constraint-based causal discovery methods, the worst-case time complexity remains exponential, which may limit its scalability for high-dimensional or long time series datasets. And, in real-world scenarios, nonstationarity may not only induce spurious dependencies but also lead to

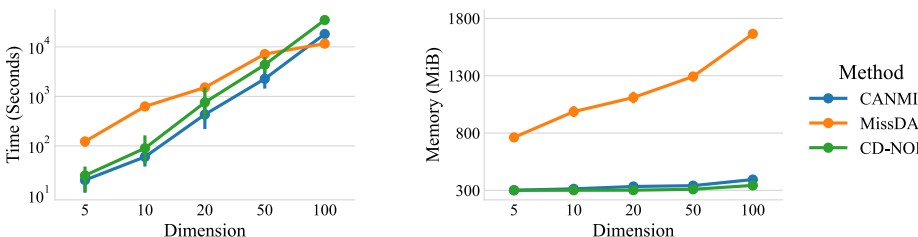

Figure 4: Comparison of computational efficiency between CANMI and its baselines with various $d \in \{5, 10, 20, 50, 100\}$.

incorrect edge orientations. It is important to investigate whether such phenomena can actually occur and, if so, to develop algorithms for such phenomena.

## G    BROADER IMPACTS

The primary objective of this work was to take a significant step toward causal discovery in more realistic and challenging conditions, specifically nonstationary and MNAR missingness, which are commonly encountered in neuroscience, healthcare, and sensor networks, as explained in the introduction. Therefore, although we believe that our method will serve beneficial purposes, it is still important for the reader to be aware of the underlying assumptions, which are relatively general and commonly used in previous causal discovery approaches and therefore are unlikely to interfere with most practical applications; however, if these assumptions are violated, misinterpretation may still lead to incorrect conclusions.

