# OpenReview forum: "CANMI: Causal Discovery under Nonstationary Missingness Mechanisms"
_ICLR.cc/2026/Conference — Submitted to ICLR 2026_

### Official Review · Reviewer_AGVp · 2025-10-30

**Soundness:** 2
**Presentation:** 2
**Contribution:** 2
**Rating:** 2
**Confidence:** 4

**Summary:**

This paper addresses the challenging problem of causal discovery from time series data with missing values, where the missingness mechanism is nonstationary. They provide theoretical guarantees for identifiability and distribution recovery, and demonstrate empirical performance on synthetic and real-world datasets, claiming superiority over existing baselines.

**Strengths:**

Strengths
- The paper tackles a practical nonstationary missingness issue, which is important in many real-world applications.

- The experiments on synthetic and real-world datasets suggest that the promising results of CANMI algorithm under the nonstationary missing data.

**Weaknesses:**

Weaknesses

- The main issue is that the contribution of this work is somewhat limited. The methodological and theoretical foundation of the work appears to be a relatively straightforward extension of MVPC [1]. Moreover, the main contribution of the nonstationary missingness mechanism is simply solved by introducing a time index, which, however, is only a direct application of [2].

- Furthermore, the definition of nonstationary that is used in this work is not formally defined. One typical type of nonstationary data should be defined on time series data, with, for example, different lengths of time lag and the Granger causality. This vagueness makes it difficult to assess the generalizability and boundaries of the proposed approach.

- The method's reliance on using time as a proxy variable is not sufficiently justified. The paper lacks a clear discussion of the assumptions and conditions under which a simple time index can adequately capture complex, underlying nonstationary processes. For example, how the time index models the time series data with a time lag.

[1] Ruibo Tu, Cheng Zhang, P Ackermann, H Kjellström, and Kun Zhang. Causal discovery in the presence of missing data. In Proceedings of the 22nd International Conference on Artificial Intelligence and Statistics, pp. 1762–1770. PMLR, 2019.

[2] Biwei Huang, Kun Zhang, Jiji Zhang, J Ramsey, Ruben Sanchez-Romero, C Glymour, and B Scholkopf. Causal discovery from heterogeneous/NOnstationary data. Journal of machine learning research, 21(89):89:1–89:53, 2020.

**Questions:**

See the weaknesses above.

---

> ### Author Response · Authors · 2025-11-21
>
> We greatly appreciate the valuable suggestions and comments on our manuscript. We will provide detailed responses below.
>
> > The main issue is that the contribution of this work is somewhat limited. The methodological and theoretical foundation of the work appears to be a relatively straightforward extension of MVPC [1]. Moreover, the main contribution of the nonstationary missingness mechanism is simply solved by introducing a time index, which, however, is only a direct application of [2].
> >
>
> Thank you for the comments. Certainly, our work is closely related to MVPC and CD-NOD. However, the resulting approach is not a simple combination of them. We focus on nonstationary missingness mechanisms, which represent a complex scenario where missing values are caused by both other variables and nonstationarity. Our main contribution is that the joint distribution and missingness mechanisms are identifiable from data with nonstationary missingness mechanisms, even when the nonstationary environment and the missingness mechanisms are mutually dependent. We then design CANMI based on these theoretical results, ensuring that our proposed algorithm is sound.
>
> > Furthermore, the definition of nonstationary that is used in this work is not formally defined. One typical type of nonstationary data should be defined on time series data, with, for example, different lengths of time lag and the Granger causality. This vagueness makes it difficult to assess the generalizability and boundaries of the proposed approach.
> >
>
> Thank you for the comments. In our work, nonstationarity is induced by time-evolving factors $U$, whose distribution is written as $P_t(U)$. Through the latent factors, we allow the data distribution to vary over time, i.e., there exist $t_1\neq t_2$ such that $P_{t_1}({V,R})\neq P_{t_2}({V,R})$. We have added this formula description of nonstationarity to Section 3.1 in the revised version.
>
> We also clarify the relationship to time-lagged dependencies. Even if time series data have time-lagged relations such as Granger causality, it is not necessarily nonstationary data. For example, a typical VAR model generally represents a stationary process. Therefore, our method does not use the time index to represent time-lagged dependencies themselves. Note that time-lagged dependencies can instead be modeled by adding lagged variables as separate nodes in the graph, and CANMI can in principle be straightforwardly applied to such an augmented graph [1]. The role of the time index in CANMI is specifically to capture nonstationary changes in the causal and missingness mechanisms, rather than to represent time-lagged dependencies.
>
> [1] T Chu and C Glymour. Search for additive nonlinear time series causal models. Journal of machine learning research: JMLR, 9:967–991, 1 June 2008.
>
> > The method's reliance on using time as a proxy variable is not sufficiently justified. The paper lacks a clear discussion of the assumptions and conditions under which a simple time index can adequately capture complex, underlying nonstationary processes.
> >
>
> Thank you for the comments. The formulation using the time index is designed to capture practically common phenomena where the dominant changes are driven by a smoothly time-varying environment, such as seasonal effects and user behavior trends. Within this modeling framework, a simple time index can capture quite complex nonstationary behavior. Specifically, since we do not impose a restrictive parametric form on how the mechanisms depend on $T$, our approach can in principle accommodate various types of complex patterns, including nonlinear and periodic behavior. Also, our empirical results on the fMRI data, where nonstationarity in neural connectivity is widely observed, provide supporting evidence that such a representation is effective in practice for this type of nonstationarity. At the same time, our formulation does not aim to cover all possible forms of nonstationarity. In particular, abrupt changes and sudden interventions that cannot be well approximated by a smooth function of the time index are outside the scope of the current framework. Extending CANMI to explicitly model such non-smooth nonstationary phenomena is an important direction for future work. We have added the above discussion to Appendix C.1 of the revised version.

---

### Official Review · Reviewer_hsKe · 2025-10-30

**Soundness:** 3
**Presentation:** 3
**Contribution:** 3
**Rating:** 6
**Confidence:** 3

**Summary:**

This paper tackles the challenging problem of causal discovery from time series data in the presence of nonstationary missingness mechanisms, including MNAR (Missing Not At Random) cases. The authors propose a constraint-based approach that extends the framework of Tu et al. (2019) by incorporating time information T to handle nonstationarity. The experimental results demonstrate the effectiveness of the proposed method.

**Strengths:**

1. The paper addresses an important and challenging problem—causal discovery under nonstationary and MNAR settings in time series data. This topic is of broad interest to both the causal discovery and time-series communities.

2. The authors provide a good review of related literature.

3. The paper is well-organized and clearly written.

**Weaknesses:**

1. The theoretical analysis focuses mainly on the non-missing case (R=0), i.e., when no missing values are present. Proposition 3 appears to closely resemble the result in Qiao et al. (2024), so the novelty of the theoretical part may be limited.

2. Assumption 1 ensures that nonstationarity can be addressed by including T as a variable. It would be helpful to discuss what happens if this assumption is violated—e.g., whether the method becomes biased or fails to identify correct causal directions.

3. The faithfulness assumption is made on the distribution P, rather than on the missing-data distribution. In principle, faithfulness should be discussed with respect to the augmented distribution including missingness indicators. If my understanding is incorrect, clarification would be appreciated.

4. The formal statements of Proposition 2 and Proposition 3 are somewhat unclear in the context of missing data. It would strengthen the paper to explicitly formulate these results using the missingness graph (m-graph) representation and clarify their implications.

Reference:

[1]. Identification of causal structure in the presence of missing data with additive noise model. AAAI 2024

**Questions:**

See Weaknesses.

---

> ### Author Response · Authors · 2025-11-21
>
> We greatly appreciate the valuable suggestions and comments on our manuscript. We will provide detailed responses below.
>
> > The theoretical analysis focuses mainly on the non-missing case (R=0), i.e., when no missing values are present.
> >
>
> Thank you for the comments. $R=0$ does not indicate the non-missing case, but show that we can only use partially observed data. Correctly, no including $R$ into the conditioning set indicates the non-missing case. Also, if data is MAR or MNAR, it does not hold $R\perp V$, then $P(V)\neq P(V\mid R=0)$, indicating that ignoring the causes of missingness leads to the biased results. For example, regarding Theorem 1, we prove that we can recover the data distribution $P(V)$, which we cannot directly obtain under nonstationary missingness mechanisms, using only the distribution of partially observed data $P(V\mid R=0)$ and parents set of each variable.
>
> > Proposition 3 appears to closely resemble the result in Qiao et al. (2024), so the novelty of the theoretical part may be limited.
> >
>
> Thank you for the comments. Certainly, Proposition 3 in our paper partly resembles the result in Qiao et al. (2024). However, we focus on nonstationary missingness mechanisms, which represent a complex scenario where missing values are caused by both other variables and nonstationarity. Our main contribution is that the joint distribution and missingness mechanisms are identifiable from data with nonstationary missingness mechanisms, even when the nonstationary environment and the missingness mechanisms are mutually dependent.
>
> > Assumption 1 ensures that nonstationarity can be addressed by including T as a variable. It would be helpful to discuss what happens if this assumption is violated—e.g., whether the method becomes biased or fails to identify correct causal directions.
> >
>
> Thank you for the comments. As you mentioned, our formulation does not aim to cover all possible forms of nonstationarity. If this assumption is violated (e.g., under abrupt changes or instantaneous shocks), the time index may fail to capture the underlying latent factors. Theoretically, this leads to unblocked back-door paths through unobserved variables, resulting in biased CI tests and the potential detection of spurious edges or incorrect orientations.
>
> > The faithfulness assumption is made on the distribution P, rather than on the missing-data distribution. In principle, faithfulness should be discussed with respect to the augmented distribution including missingness indicators.
> >
>
> Thank you for the comments. As you mentioned, the faithfulness assumption should be defined over the augmented distribution including missingness indicators. In fact, all theoretical results and our method are derived based on such assumptions.
>
> > The formal statements of Proposition 2 and Proposition 3 are somewhat unclear in the context of missing data. It would strengthen the paper to explicitly formulate these results using the missingness graph (m-graph) representation and clarify their implications.
> >
>
> Thank you for the suggestions. Proposition 2 and Proposition 3 are not related to overall m-graph, but to each local conditional independence in dataset. Therefore, in the revised versions, we have partially revised their surrounding descriptions to clarify their interpretation in terms of m-graph.

---

### Official Review · Reviewer_ae6Y · 2025-10-31

**Soundness:** 3
**Presentation:** 1
**Contribution:** 2
**Rating:** 4
**Confidence:** 3

**Summary:**

This paper tackles an important and underexplored problem in causal discovery—handling nonstationary missingness mechanisms (i.e., missingness depending on both other variables and time variation). The authors propose CANMI, a constraint-based algorithm capable of identifying causal structures using only partially observed time series data, even under MNAR settings. The work provides theoretical guarantees (identifiability, recoverability, and order independence) and validates the method on both synthetic and real-world fMRI datasets.

**Strengths:**

The problem stated in the paper is interesting and important. The paper provides theoretical guarantees about identifiability.

**Weaknesses:**

The overall presentation of the paper requires substantial improvement. For instance, the paper currently lacks a **problem setup section**, which should clearly introduce the problem formulation and provide concrete examples of scenarios that fit the proposed framework. In addition, Section 3.1 asserts that nonstationarity and missingness can induce spurious causal relations; it would be clearer and more rigorous to formalize this claim as a **proposition** and illustrate it with specific examples. Furthermore, the **role of Step 1** in the algorithm remains unclear. Once ( P(V) ) can be recovered, the causal structure can, in principle, be identified. According to Theorem 1, it is sufficient to determine the **parent set of each missingness indicator**, suggesting that Steps 2, 4, and 5 alone may suffice for causal recovery.

**Questions:**

See the weakness above.

---

> ### Author Response · Authors · 2025-11-21
>
> We greatly appreciate the valuable suggestions and comments on our manuscript. We will provide detailed responses below.
>
> > The overall presentation of the paper requires substantial improvement. For instance, the paper currently lacks a **problem setup section**, which should clearly introduce the problem formulation and provide concrete examples of scenarios that fit the proposed framework. In addition, Section 3.1 asserts that nonstationarity and missingness can induce spurious causal relations; it would be clearer and more rigorous to formalize this claim as a **proposition** and illustrate it with specific examples.
> >
>
> Thank you for the suggestions. In the revised manuscript, we have substantially rewritten Section 3.1 so that it explicitly serves as the problem setup section, and we have introduced Proposition 1 in Section 3.1 together with a simple causal-graph example in its proof to illustrate how nonstationary missingness mechanisms can induce spurious causal relations and to describe our problem setting more clearly.
>
> > Furthermore, the **role of Step 1** in the algorithm remains unclear. Once ( P(V) ) can be recovered, the causal structure can, in principle, be identified. According to Theorem 1, it is sufficient to determine the **parent set of each missingness indicator**, suggesting that Steps 2, 4, and 5 alone may suffice for causal recovery.
> >
>
> Thank you for the comments. The output of Step 1 is used in later steps. Specifically, by identifying which variables have time-variant causal mechanisms in Step 1, we can determine whether the time index should be included in the conditioning set. This allows us to avoid unnecessary conditioning, thereby improving both the statistical power of the CI tests and the computational efficiency.

---

### Official Review · Reviewer_FSq5 · 2025-11-01

**Soundness:** 3
**Presentation:** 2
**Contribution:** 3
**Rating:** 4
**Confidence:** 5

**Summary:**

The paper proposes a constraint-based algorithm for discovering causal relations in time-series data that handles nonstationary mechanisms and randomly missing data by introducing missingness indicators and employing importance resampling. The authors claim it is the first method to jointly address these two challenges in this setting. Experiments on synthetic and real-world datasets show consistently higher accuracy than baseline methods, and the reported runtime is comparable to prior work.

**Strengths:**

1. The paper is complete, providing both theoretical guarantees and a series of experiments under various settings.

2. Empirically, the algorithm shows higher accuracy than baseline methods over various settings.

3. The work tackles two practical challenges: nonstationarity and missing data, both common in real-world settings.

4. The use of joint-distribution reconstruction and importance resampling to reduce the false positive rate under nonstationarity and missingness is interesting and novel.

**Weaknesses:**

1. The paper’s main novelty lies in Theorem 1 (Step 4), whereas Propositions 1–3 and Steps 1–3 are relatively standard under Assumption 5 after introducing missing indicators. The central challenge in time series, exacerbated by nonstationarity and missing data, is spurious (false-positive) edges. Therefore, the parts on distribution reconstruction and importance resampling should be explained more clearly and given greater emphasis. For example, Step 2 is not directly useful for causal discovery; it serves as a prerequisite for Step 4 to remove spurious edges. Additionally, is the output of Step 1 used in later steps? (See the Questions section for further points about Step 1.) Given this, an ablation study with and without Step 4 would be valuable, since Step 4 appears to be the key step.

2. The rationale for the chosen baselines is unclear, since many of them are not designed for time series. Why use LiNGAM instead of VAR-LiNGAM, NOTEARS instead of DYNOTEARS, and why omit PCMCI, which is widely used as a time-series causal discovery baseline?

3. While the proposed algorithm can identify variables with time-invariant mechanisms, can it also identify variables whose mechanisms are time-variant correctly? Step 1 is titled Detecting changing causal mechanisms, but it discusses only invariance. Could edges between variables and $T$ also arise from spurious correlations? If the method can detect variables with changing mechanisms, it would be helpful to report the accuracy for this sub-task.

4. It would be better to include an official computational analysis.

Other comment:

Consider adding an assumption that nonstationarity changes effect sizes only, while edge presence remains fixed.

**Questions:**

1. What is the difference between Step 1 and CD-NOD? Can CD-NOD be directly extended to the missing-data setting by combining distribution reconstruction and importance resampling?

2. Beyond nonstationarity and missing data, spurious edges can occur even in stationary time series without missing data due to autocorrelation. Can Step 4 also address it?

3. After Step 4, are all obtained edges retested on the modified datasets? If spurious edges arise between observed variables and missingness indicators in earlier steps, and hence the indicator’s parent set is Eq.3 is a superset of the true parents, what is the impact?

4. Which conditional independence tests are used in the proposed algorithm, given that $R$ is binary while other variables may be continuous?

5. For the baselines, do you use all $N$ samples or only complete cases after listwise deletion?

6. The learned causal graph is a summary graph that includes only lag 0 effects. How is the comparison performed when a baseline returns a graph with time lags? What alignment and scoring procedure do you use across lags? Are any baselines used that do not allow contemporaneous edges?

---

> ### Author Response · Authors · 2025-11-21
>
> We greatly appreciate the valuable suggestions and comments on our manuscript. We will provide detailed responses below.
>
> > What is the difference between Step 1 and CD-NOD? Can CD-NOD be directly extended to the missing-data setting by combining distribution reconstruction and importance resampling?
> >
>
> Thank you for the question. In fact, once the overall data distribution $P(V)$ is recovered according Theorem 1 and generate a modified dataset by importance resampling, CD-NOD can be applied on these samples. However, first, recovering $P(V)$ under nonstationary missingness mechanisms is main contribution of our work. Additionally, our method is designed to be more sample-efficient than a direct application CD-NOD on recovered data. Step 3 first removes edges that are guaranteed to be absent in an underlying true m-graph based only on CI test results in the partially observed data (Proposition 3), and Step 4 uses distribution reconstruction and importance resampling only to prune potential spurious edges induced by the nonstationary missingness mechanisms. This two-stage design is specific to the missing-data setting and goes beyond simply plugging CD-NOD into a preprocessed dataset.
>
> > Beyond nonstationarity and missing data, spurious edges can occur even in stationary time series without missing data due to autocorrelation. Can Step 4 also address it?
> >
>
> Thank you for the question. Our method addresses temporal dependencies, including autocorrelation, in Step 1 of our algorithm. Specifically, for each variable $X_i$, we perform CI tests with respect to the time index $T$, conditioned on subsets of the remaining variables. If $X_i$ is found to be conditionally independent of $T$ given some subset $S \subseteq V \setminus \{X_i\}$, we regard its causal mechanism as time-invariant. Otherwise, it is treated as time-variant. This allows us to detect and control for latent time-varying confounders before conducting subsequent CI tests. By doing so, we mitigate spurious dependencies due to autocorrelation or latent temporal patterns.
>
> Additionally, CANMI can handle autocorrelations induced by time-varying causal mechanisms, but autocorrelations occurred by lagged variables are outside the scope of our work. However, in practice, time-lagged dependencies can instead be modeled by adding lagged variables as separate nodes in the graph, and CANMI can in principle be straightforwardly applied to such an augmented graph [1]. In particular, if we use MCI tests [2] for time-lagged dependencies, CANMI also spurious edges due to autocorrelations occurred by lagged variables.
>
> [1] T Chu and C Glymour. Search for additive nonlinear time series causal models. Journal of machine learning research: JMLR, 9:967–991, 1 June 2008.
>
> [2] Jakob Runge, Peer Nowack, Marlene Kretschmer, Seth Flaxman, and Dino Sejdinovic. Detecting and quantifying causal associations in large nonlinear time series datasets. Science advances, 5 (11):eaau4996, November 2019.
>
> > After Step 4, are all obtained edges retested on the modified datasets?
> >
>
> Thank you for the question. In Proposition 3, we derive the situations in which a causal dependency between $X$ and $Y$ in the observed data may be extraneous. Based on this, we identify a subset of edges obtained in Step 3 as candidates for being spurious. Therefore, we do not retest all obtained edges, but only those candidate edges that satisfy the conditions in Proposition 3.
>
> > Which conditional independence tests are used in the proposed algorithm, given that $R$ is binary while other variables may be continuous?
> >
>
> Thank you for the question. All CI relations in the proposed algorithm are tested using the kernel-based conditional independence (KCI) test. This is a nonparametric test that does not assume any specific parametric form for the joint distribution.
>
> > For the baselines, do you use all $N$ samples or only complete cases after listwise deletion?
> >
>
> Thank you for the question. For the baselines which cannot handle missing values, we used the observed data in which missing values are imputed using Zero-order Holder.
>
> > The learned causal graph is a summary graph that includes only lag 0 effects. How is the comparison performed when a baseline returns a graph with time lags? What alignment and scoring procedure do you use across lags? Are any baselines used that do not allow contemporaneous edges?
> >
>
> Thank you for the questions. Some baselines return a causal graph with time lags. We convert such a graph into a summary graph for fair comparison. Specifically, we first obtain a causal graph with no time lags. Then, for each time lag, we add an time-lagged dependency while a resulting causal graph is acyclic. Additionally, no baseline methods only return time-lagged effects without contemporaneous edges.

---

> ### Author Response · Authors · 2025-11-21
>
> > If spurious edges arise between observed variables and missingness indicators in earlier steps, and hence the indicator’s parent set is Eq.3 is a superset of the true parents, what is the impact?
> >
>
> Thank you for the question. We prove that the parents of missingness indicators are identifiable in Proposition 1. So, theoretically, the indicator’s parent set in Eq.3 does not become a superset of the true parents. However, in practice with finite samples, the estimated parent set may indeed become a superset of the true parents. In such cases, the validity of the density ratio estimation (Theorem 1) holds because conditioning on the extra variables does not lead to bias. The primary impact of including extraneous parents is a potential increase in the variance of the estimated weights, which could slightly reduce the statistical power of the subsequent CI tests in Step 4. However, this is preferable to omitting true parents, which would lead to bias.
>
> > The central challenge in time series, exacerbated by nonstationarity and missing data, is spurious (false-positive) edges. Therefore, the parts on distribution reconstruction and importance resampling should be explained more clearly and given greater emphasis.
> >
>
> Thank you for the comments. As you pointed out, the main contribution of our work is to address spurious edges under nonstationary missingness mechanisms. To emphasize this, we have updated the introduction in the revised version. Additionally, we state in Section 3.3 that although Step 2 is not directly useful for causal discovery, it is important for our algorithm to recover the data distribution.
>
> > Additionally, is the output of Step 1 used in later steps? (See the Questions section for further points about Step 1.)
> >
>
> Yes, the output of Step 1 is used in later steps. Specifically, by identifying which variables have time-variant causal mechanisms in Step 1, we can determine whether the time index should be included in the conditioning set. This allows us to avoid unnecessary conditioning, thereby improving both the statistical power of the CI tests and the computational efficiency.
>
> > Given this, an ablation study with and without Step 4 would be valuable, since Step 4 appears to be the key step.
> >
>
> Thank you for the invaluable suggestions of an ablation study. We prepared a limited version, namely CANMI-L, which leaves extraneous edges without Step 4. The following Table shows the ablation study result.
>
> | Method | Graph | TPR | FPR | FDR | F1 Score | SHD |
> | --- | --- | --- | --- | --- | --- | --- |
> | CANMI-L | ER2 | 0.887 ± 0.151 | 0.079 ± 0.043 | 0.220 ± 0.098 | 0.825 ± 0.103 | 3.0 ± 1.717 |
> | CANMI-L | ER4 | 0.726 ± 0.127  | 0.071 ± 0.033  | 0.140 ± 0.062  | 0.783 ± 0.096  | 5.2 ± 2.546 |
> | CANMI | ER2 | 0.922 ± 0.130  | 0.029 ± 0.016  | 0.093 ± 0.062  | 0.910 ± 0.083  | 1.6 ± 1.517 |
> | CANMI | ER4 | 0.765 ± 0.160  | 0.043 ± 0.047  | 0.092 ± 0.115  | 0.822 ± 0.120  | 4.6 ± 2.608 |
>
> We can see that CANMI-L causes a drop in accuracy, suggesting that the recovery of the data distribution are crucial for reliable causal discovery under nonstationary missingness mechanisms and this results aligns with the discussion presented in Theorem 2. We have incorporated this result into Table 2 and the above discussion into the revised version.
>
> > The rationale for the chosen baselines is unclear, since many of them are not designed for time series. Why use LiNGAM instead of VAR-LiNGAM, NOTEARS instead of DYNOTEARS, and why omit PCMCI, which is widely used as a time-series causal discovery baseline?
> >
>
> Thank you for the question. Certainly, there are some existing methods designed for time series data, such as VAR-LiNGAM, DYNOTERS, and PCMCI. They can obtain lagged relations, but they implicitly assume the stationary data. In contrast, we focus on nonstationary changes in the causal and missingness mechanisms, where standard stationarity-based methods may struggle. Additionally, time-lagged dependencies can be represented by augmenting the variable set with lagged variables and CANMI can be applied to such an augmented graph as discussed above.
>
> > While the proposed algorithm can identify variables with time-invariant mechanisms, can it also identify variables whose mechanisms are time-variant correctly? Step 1 is titled Detecting changing causal mechanisms, but it discusses only invariance.
> >
>
> Thank you for this comment. Our algorithm distinguishes time-variant mechanisms: in Step 1, a variable $X$ is classified as time-invariant if there exists a conditioning set $S \subseteq V \setminus {X}$ such that $X \perp T \mid S$; otherwise, $X$ is treated as having a time-varying mechanism. Under our structural assumptions and faithfulness, this existence/non-existence of such an $S$ is equivalent to the mechanism of $X$ being time-invariant/variant. We have clarified this point in the revised manuscript.

---

### Author Response · Authors · 2025-12-01
**Summary for the New Area Chairs**

We would like to thank the Area Chairs for handling this paper under challenging circumstances following the recent incident. Below, we provide a concise digest of the key information about our work.

First, we briefly summarize **the main strengths of our work**.

1. **Important problem (All Reviewers)**: To the best of our knowledge, this is the first work to investigate the challenges associated with causal discovery in the complex scenarios where distributional nonstationarity and missingness mechanisms are mutually dependent (i.e., nonstationary missingness mechanisms). Such scenarios are ubiquitous in various applications, including sensor networks and brain signal data, and they reflect the complexity of real-world conditions.
2. **Theoretical foundations (Reviewers FSq5 and ae6Y)**: Our problem is important, but it is non-trivial because we only have access to partially observed data whose distribution is distorted by both nonstationarity and non-random missingness, relative to the complete data distribution. This distortion may lead to the unintentional detection of spurious edges and the omission of correct ones. To overcome the difficulty, we established a set of theoretical results, showing that our algorithm recovers the correct causal structure.
3. **Experiments under various settings (Reviewers FSq5 and AGVp)**: We conducted extensive experiments on both synthetic and real-world datasets. Our experimental results in Figure 1 show that our algorithm achieved consistent performance even in the presence of substantial missing values, whereas baseline methods exhibited degraded accuracy as the missingness increased.

Next, we would like to highlight **the major revisions** made in response to the reviewers’ comments.

1. **Clarify the problem definition**: We have revised Section 3.1, "Problem Definition," to more clearly describe the challenges addressed in our work. In particular, we have introduced Proposition 1 which formally characterizes the discrepancy induced by nonstationary missingness mechanisms.
2. **Conduct additional experiments**: We prepared a limited version, namely CANMI-L, which leaves extraneous edges unpruned without Step 4. The following table shows the ablation study results. We can see that CANMI-L causes a drop in accuracy, suggesting that the recovery of the data distribution is crucial for reliable causal discovery under nonstationary missingness mechanisms and these results align with the discussion presented in Theorem 1. In particular, FDR and SHD values on ER2 graphs were significantly higher. This is because more extraneous edges were generated due to the sparsity of the true causal graph, and CANMI-L failed to prune them.


    | Method | Graph | TPR | FPR | FDR | F1 Score | SHD |
    | --- | --- | --- | --- | --- | --- | --- |
    | CANMI-L | ER2 | 0.887 ± 0.151 | 0.079 ± 0.043 | 0.220 ± 0.098 | 0.825 ± 0.103 | 3.0 ± 1.717 |
    | CANMI-L | ER4 | 0.726 ± 0.127  | 0.071 ± 0.033  | 0.140 ± 0.062  | 0.783 ± 0.096  | 5.2 ± 2.546 |
    | CANMI | ER2 | 0.922 ± 0.130  | 0.029 ± 0.016  | 0.093 ± 0.062  | 0.910 ± 0.083  | 1.6 ± 1.517 |
    | CANMI | ER4 | 0.765 ± 0.160  | 0.043 ± 0.047  | 0.092 ± 0.115  | 0.822 ± 0.120  | 4.6 ± 2.608 |
3. **Clarify the differences between our algorithm and stationary time series causal discovery algorithms**: Our work primarily focuses on nonstationary data, rather than on time-lagged dependencies. Stationary time series causal discovery methods usually assume a time-invariant data-generating process. In addition, time-lagged dependencies can instead be modeled by adding lagged variables as separate nodes in the graph, and our algorithm can in principle be straightforwardly applied to such an augmented graph. Therefore, the main distinction lies in our emphasis on handling complex nonstationary missingness in the contemporaneous causal structure, and our approach is in this sense complementary to existing stationary time series causal discovery algorithms.

Thanks to the insightful comments from the reviewers, we have been able to further improve our work. We would like to express our gratitude to all reviewers for the time and effort spent reviewing our manuscript. We believe the revised version satisfactorily addresses reviewers’ concerns.

---

### Meta-Review · Area_Chair_DXSB · 2026-01-06

**Summary:**

Reviewer FSq5 raised several good points, the most important of which is that the paper, claiming to propose a new causal discovery algorithm in time series data, does NOT compare with well-established baselines for time-series causal discovery. This is a dealbreaker, and I am surprised to see the authors in their rebuttal justify this decision instead of providing empirical comparisons.

Reviewer ae6Y raised issues about the clarity of presentation.

Reviewer hsKe is supportive of acceptance, but based on the important points he/she raised, I believe the score should have been lower in the rejection category. This reviewer brought up technical issues, some of which are addressed by the authors, but some require updates to the manuscript.

Reviewer AGVp had issues with the novelty of the work.

**Reviewer Concerns:**

Reviewer FSq5's points about the lack of relevant time-series baselines is not addressed by the authors.

It is very hard to address the novelty concern of Reviewer AGVp so I think this is still outstanding.

**Reviewer Scores:**

Reviewer FSq5 would not have changed their score based on the above point.

Reviewer ae6Y would not have been satisfied because they probably would have needed to see a new manuscript.

Reviewer hsKe may have reduced their score based on other reviews, but I am not very sure about this. They certainly would not further increase their score.

Reviewer AGVp would not have changed their score based on rebuttal in my opinion.

---

### Decision · Program_Chairs · 2026-01-26

Reject